# Neocortical inhibitory interneuron subtypes are differentially attuned to synchrony- and rate-coded information

Luke Y. Prince [1,2,3,8], Matthew M. Tran[3,4,8], Dorian Grey[3], Lydia Saad[3], Helen Chasiotis[3], Jeehyun Kwag[5], Michael M. Kohl [6] & Blake A. Richards [1,2,4,7 ✉]

Neurons can carry information with both the synchrony and rate of their spikes. However, it is unknown whether distinct subtypes of neurons are more sensitive to information carried by synchrony versus rate, or vice versa. Here, we address this question using patterned optical stimulation in slices of somatosensory cortex from mouse lines labelling fast-spiking (FS) and regular-spiking (RS) interneurons. We used optical stimulation in layer 2/3 to encode a 1-bit signal using either the synchrony or rate of activity. We then examined the mutual information between this signal and the interneuron responses. We found that for a synchrony encoding, FS interneurons carried more information in the first five milliseconds, while both interneuron subtypes carried more information than excitatory neurons in later responses. For a rate encoding, we found that RS interneurons carried more information after several milliseconds. These data demonstrate that distinct interneuron subtypes in the neocortex have distinct sensitivities to synchrony versus rate codes.

[1] Mila - Quebec Artificial Intelligence Institute, Montreal, QC, Canada. [2] School of Computer Science, McGill University, Montreal, QC, Canada. [3] Department of Biological Sciences, University of Toronto Scarborough, Toronto, ON, Canada. [4] Department of Cell and Systems Biology, University of Toronto, Toronto, ON, Canada. [5] Department of Brain and Cognitive Engineering, Korea University, Seoul, South Korea. [6] Institute of Neuroscience and Psychology, University of Glasgow, Glasgow, UK. [7] Department of Neurology and Neurosurgery, McGill University, Montreal, QC, Canada. [8]These authors contributed equally: Luke Y. Prince, Matthew M. Tran. ✉email: blake.richards@mila.quebec

One of the foundational concepts in neuroscience is that neurons encode information with their action potentials. The earliest demonstrations that action potentials carry information came from Lord Adrian (1926), who demonstrated that the rate of spikes in peripheral nerves is correlated with the force applied to a limb[1]. Decades of work following this demonstrated that the rate of spikes can carry information about almost any aspect of the environment or an animal's behaviour, including sensory stimuli[2], spatial location[3], action selection[4], etc. However, the rate-of-fire of a neuron is not the only aspect of a spike train that can carry information[5]. It has also been demonstrated that the specific timing of action potentials can correlate with salient variables, including sensory stimuli[6,7], spatial location[8], and action selection[9,10]. There are a variety of potential coding schemes using spike times[5], but basic principles of postsynaptic spatiotemporal integration tell us that the synchrony of incoming inputs can be as important as the rate of incoming inputs to a neuron[11]. Therefore, the brain is likely to encode information using both the synchrony and rate of spikes[12–14].

One interesting aspect of information encoding with both synchrony and rate of spikes is that different cells with different biophysical properties will respond to each signal differently[15,16]. For example, when we consider linear integration, a cell with a short membrane time constant will be more sensitive to information carried by synchronous inputs, whereas a cell with a longer time constant would be more sensitive to the rate of inputs. These issues are particularly salient when we consider the diversity of biophysical properties found in neocortical inhibitory interneurons[17–19]. Different types of inhibitory interneurons possess intrinsic membrane properties, morphologies, firing patterns, and presynaptic inputs[17,19]. For example, inhibitory interneurons that exhibit fast-spiking (FS) behaviour are known to have very short membrane time constants, short-term depressing presynaptic inputs and often express parvalbumin (PV). In contrast, inhibitory interneurons that show regular-spiking (RS) patterns with spike-frequency adaptation also display short-term facilitating presynaptic inputs and typically express somatostatin (SST). These distinct biophysical properties of interneurons are likely relevant to information encoding in the brain[16,20,21].

Given that different functional subtypes of interneurons, such as FS and RS cells, display distinct biophysical properties, it is possible that each subtype is more or less sensitive to information conveyed by synchrony or rate. However, although these interneuron subtypes have been studied extensively over recent decades, it is unknown whether there are functional specializations in the integration of information carried by synchrony or rate of activity.

Here, we explored this question using a combination of transgenics, ex vivo whole-cell patch clamping, and patterned optogenetic illumination. Specifically, we examined the responses of green-fluorescent protein positive (GFP+) neurons in two different transgenic mouse lines: GAD67-GFP (reported to target FS interneurons) and GIN-GFP (reported to target RS SST+ interneurons). We recorded from GFP+ and GFP− (likely excitatory neurons) in layer 2/3 of mouse barrel cortex. We then activated the cells optogenetically in a patterned manner. Using a digital micromirror device, we encoded a random 1-bit signal by controlling either the synchrony or rate of optical activation in the tissue. We then examined the amount of information that GAD67+, GIN+, and GFP− (likely pyramidal) neurons carried about this 1-bit signal in their spiking and membrane voltage. We found that the two classes of interneurons were differentially sensitive to the synchrony and rate of optical activation. Specifically, we observed that both interneuron types carried more

information in their spiking responses than pyramidal neurons but in different ways. FS GAD67+ interneurons carried more information about the 1-bit signal in their early (<5 ms) responses to synchronous activation. Both interneuron types carried more information than GFP− cells in response to synchronous activation after more time (>5 ms). When we examined the responses to rate encoding, RS GIN+ interneurons carried more information about the 1-bit signal than either FS GAD67+ or GFP− pyramidal neurons in their later responses. These data confirm that different types of inhibitory interneurons can integrate information carried by synchrony or rate of activity in different ways. It also suggests that the inhibition received by pyramidal neurons in the neocortex may be selectively driven by the synchrony and the rate of action potentials. This may be critical for understanding how different pieces of information are encoded and relayed in the neocortex.

## Results

**Transgenic targeting of FS and RS interneurons.** We focused our investigation on layer 2/3 (L2/3) of the somatosensory cortex (specifically barrel cortex), as this is a region of the neocortex where synchrony and rate codes have been extensively explored. In order to target different populations of interneurons within L2/3 of the barrel cortex, we used GAD67-GFP (GAD67+) and GIN-GFP (GIN+) transgenic mice which have been reported to express GFP in inhibitory interneuron types with distinct firing characteristics (FS and RS, respectively)[22,23]. To confirm this distinction of electrophysiological phenotypes, we conducted whole-cell patch-clamp recordings from GFP+ cells in GAD67-GFP and GIN-GFP mice and analysed the neurons' responses to current injection in current clamp.

In the GAD67-GFP transgenic line, nearly all of the GFP+ cells ($N = 26$) exhibited spiking behaviour typical of cortical FS cells with high f–I slopes and close to linear relationships between injected current and firing frequency in each cell (Fig. 1a–c), as shown previously[23]. GFP+ cells in the GIN-GFP transgenic line ($n = 30$) showed different spiking characteristics to the GAD67+ cells as their spiking tended to accommodate in response to increasing current injections and often peaked at around 100 Hz (Fig. 1a–c), a hallmark firing pattern for almost 90% of SST+ cells[24].

To provide an additional comparison with these cell populations, we also recorded from GFP− cells in each transgenic line ($n = 30$). Usually, these GFP− cells exhibited regular firing patterns typical of layer 2/3 pyramidal cells (Fig. 1a–c)[25]. Sometimes GFP- cells, exhibited fast-spiking behaviour (4 cells), indicating that these were in fact FS interneurons, which is to be expected from a random sampling of cells in layer 2/3 of somatosensory cortex[26,27]. These four cells are shown here, but were excluded from later analyses in order to keep each population of cells distinct.

A principal components analysis (PCA) of each cell's electrophysiological characteristics (e.g. adaptation ratio, sag amplitude, and membrane tau; see 'Methods', Fig. S1) revealed three distinct clusters in the first two principal components (Fig. 1d). When these components were used to map the recorded cells onto a dendrogram, three distinct clusters that were largely consistent with selective labelling of distinct neuronal subtypes were revealed (Fig. 1e). Thus, the electrophysiological clustering supported the idea that the cells we recorded from GAD67-GFP and GIN-GFP corresponded to distinct inhibitory interneuron subtypes.

Additionally, we wanted to explore what the first few principal components corresponded to. We noted that the two populations of GFP+ interneurons could be entirely separated using only the

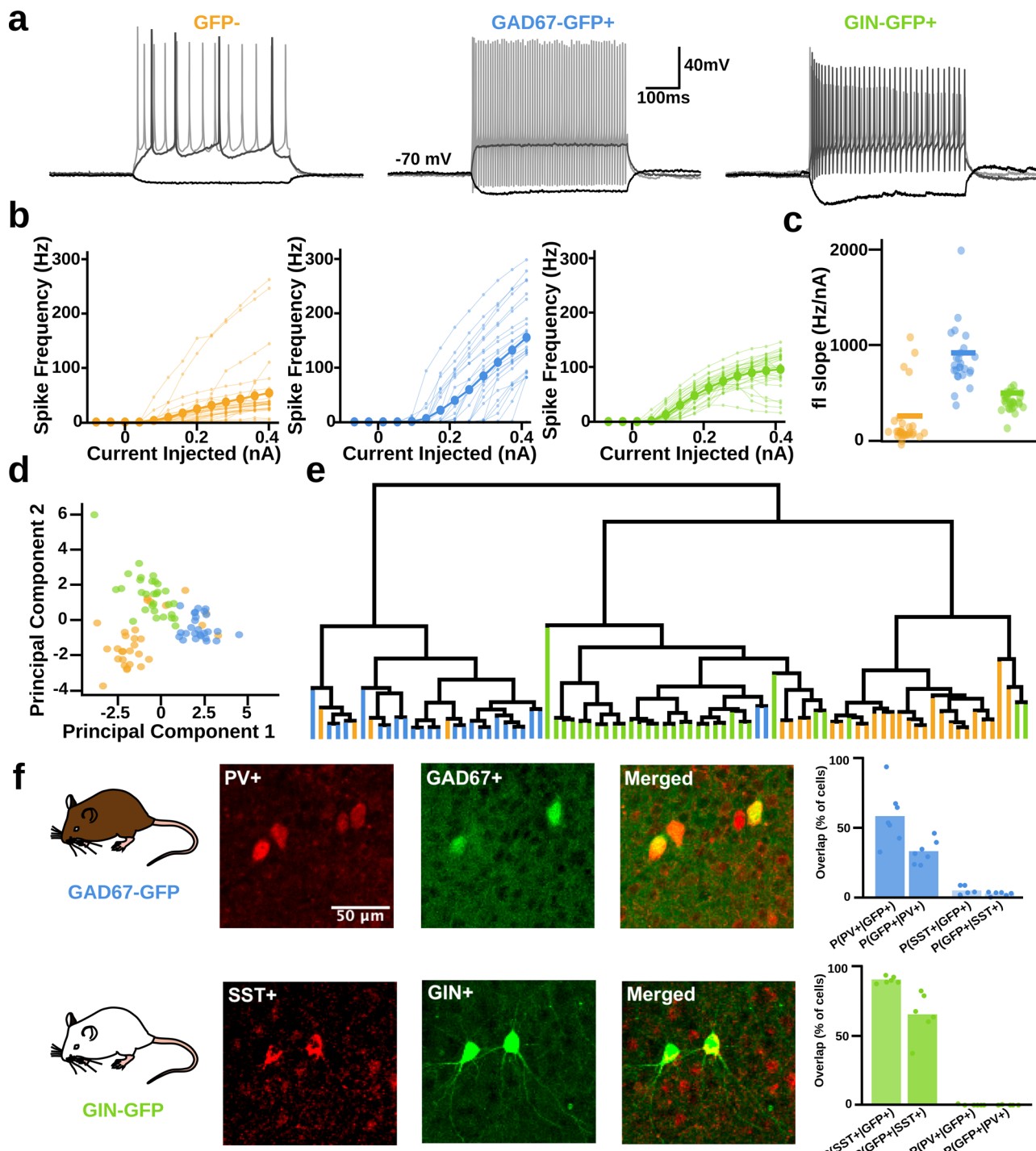

**Fig. 1 GAD67-GFP and GIN-GFP mice target distinct populations of neurons, FS, mostly PV+, and SST+ interneurons, respectively. a** Sample electrophysiological traces from each cell type to current injections −80 pA, +160 pA, +400 pA (darkest to lightest). **b** Frequency vs. current injection (f–I) curves for each cell type. Larger circles represent mean values. **c** Mean slope of the f–I curves for each cell type. Horizontal lines represent mean values. **d** 2 Component PCA of each cell type's electrophysiological characteristics. **e** Dendrogram clustering. **f** Sample immunohistochemistry images from both transgenic lines.

first principal component. The loadings for the first principal component suggest that this dimension mainly captured the linear filtering and fast-spiking properties of cells ([feature: loading]; f–I slope: 0.46, membrane time constant ($\tau$): −0.46, spike halfwidth: −0.43, rheobase: 0.32, cell capacitance: −0.30, spike amplitude: −0.23, spike threshold: −0.22, input resistance: −0.22, adaptation ratio: 0.21, sag amplitude: −0.06). In other words, cells further along the first principal component tend to

have higher f–I slope, lower membrane time constant, faster spikes, and higher rheobase. This suggests that the first principal component was largely identifying the difference between FS and RS interneurons.

The second principal component also separated GAD67+ and GIN+ cells, though not as cleanly, as there was some overlap along this dimension. The loadings for the second principal component suggested that it mainly captured the linear and non-

linear filtering properties of cells ([feature: loading]; input resistance: 0.49; cell capacitance: −0.45; sag amplitude: 0.44; adaptation ratio: 0.36; rheobase: −0.34; spike amplitude: −0.32; membrane time constant: −0.09; f–I slope: .09; spike threshold: −0.09; spike halfwidth: −0.02). This means cells further along the second principal component tended to have higher input resistance, lower cell capacitance, higher sag amplitude and higher adaptation ratios, which are features more typical of RS than FS interneurons.

For comparison with other studies, we examined co-expression of parvalbumin (PV) and somatostatin (SST) in GAD67+ and GIN+ cells using immunohistochemistry staining. Immunohistochemistry staining for PV protein in the barrel cortex of GAD-67-GFP animals showed a higher probability of GFP+ cells being PV+ (P(PV+|GFP+), 59.4% ± 7.5%, $n = 7$ animals) and vice versa (P(GFP+|PV+), 33.5% ± 3.1%, $n = 7$) when compared to staining for SST peptide (P(GFP+|SST+), 5.0% ± 1.4%, $n = 6$; P (SST+|GFP+), 3.0% ± 0.6%, $n = 6$; Fig. 1f top row). In GIN-GFP animals, the opposite was seen. In particular, when staining for SST, GIN-GFP animals showed a higher probability of GFP+ cells being SST+ (P(SST+|GFP+), 93.8% ± 1.0%, $n = 6$) and vice versa (P(GFP+|SST+), 67.6% ± 6.8%), than when compared to PV protein staining (P(PV+|GFP+), 0.2% ± 0.2% $n = 6$, P(GFP+|PV+), 0.1% ± 0.1, $n = 6$; Fig. 1, bottom row).

These results indicate that our GIN-GFP animals were expressing GFP primarily in SST+ interneurons, which suggests that our data can be directly compared with results in other studies examining SST+ cells. These results also indicate that cells recorded in GAD67-GFP animals were comprised mostly of FS PV+ interneurons. However, there are still a small number of SST+ interneurons, and interneurons expressing neither PV or SST. As was the case in previous studies[23], all GAD-67+ cells we recorded from were fast-spiking. This might mean we recorded from FS SST+ (e.g. SST + basket cells[28]), and some other FS interneurons expressing neither PV or SST (e.g. non-PV expressing L2 chandelier cells[29]). An alternative explanation is that despite colocalization with other indicators there is a selection bias towards successful patching of PV+ basket cells that tend to be larger than SST+ interneurons and have stronger GFP expression[23]. These results suggest a reasonable level of both genotypic specificity and efficiency and is in line with previous work which suggests that GAD67-GFP and GIN-GFP animals express GFP in FS and SST + cells, respectively[22–24,30].

Altogether, these results confirm that GAD67-GFP and GIN-GFP transgenic mice express GFP in cell types with distinct biophysical characteristics, corresponding to FS, mostly PV+, and RS SST+ interneurons and that most GFP− cells are not from these subclasses of interneurons. For clarity, cell groups will be labelled as GFP− (for nonfluorescent, non FS neurons), GAD67+ (for FS, likely PV+ interneurons), GIN+ (for SST+ interneurons) herein.

**Encoding a random 1-bit signal using patterned optical stimulation**. Central to our experimental objective is the ability to encode information via the synchrony or rate of activity in an ex vivo slice. To do this, we adopted a patterned optical stimulation approach. We infected the barrel cortex of 5–7-week old mice with an adeno-associated virus carrying channelrhodopsin-2 (ChR2) and the mCherry reporter, under the $Ca^{2+}$/calmodulin-dependent protein kinase II promoter (rAAV1-CamKii-hChR2 (h134r)-mCherry). This led to expression of mCherry in many neurons of layer 2/3 (Fig. 2a). Due to the use of the CamKii promoter expression of mCherry was strongest in GFP−, excitatory cells, but we observed ChR2 responses in both mCherry+

and mCherry− cells (Fig. S3), which is in-line with reports that a shortened CamKii promoter can sometimes lead to small amounts of expression beyond pyramidal neurons[31]. After sufficient time for expression (2–3 weeks), we then prepared ex vivo slices of barrel cortex and used a digital micromirror device to illuminate layer 2/3 with spatially controlled patterns of light (Fig. 2b; see 'Methods'). Whole-cell recordings from infected neurons demonstrated that we could use spatially restricted discs of illumination (470 nm, 15 µm diameter, ~14 mW/mm$^2$) to reliably induce spiking in the tissue (Fig. 2c). We also examined the responses of neurons at different distances from the disc of illumination (Fig. S2a). We found that due to the limitations of 1-photon excitation, neurons up to 50 µm away from a disc of illumination could also spike (Fig. S2b–d). Nonetheless, neurons farther than 50 µm away rarely spiked (Fig. S2b–d). To limit the spread of excitation and mixing of responses in the tissue, we chose to restrict illumination to more sparse regions of infection (see 'Methods'). Altogether, this demonstrated that we could use the digital micromirror device to activate particular regions of interest (ROIs) in the slice individually.

Next, we performed whole-cell patch-clamp recordings of GFP+ (and some GFP−) neurons in the slices. We used our patterned optical illumination approach to encode a 1-bit random signal (i.e. a signal with two states, 0 or 1) in the activity of 10 ROIs containing ChR2 expressing neurons (Fig. 3a). We chose a 1-bit random signal because it enables unbiased, low-variance estimation of mutual information with limited samples, unlike more natural, continuous signals[32]. We encoded this signal using either the synchrony or rate of optical activation of the 10 ROIs. Although, in general, neurons can and do transmit information much richer than single-bit codes[33,34], this choice of code to embed in the pattern of tissue stimulation enabled tractable estimation and analysis of mutual information in the neurons' responses (see 'Methods').

To encode the 1-bit signal using the synchrony of activation, we created an optical activation pattern where each ROI was activated at a constant rate of 2.7 Hz, but with low synchrony for the 0 state, and high synchrony for the 1 state (Fig. 3b). Specifically, during the low synchrony state, the activation times of the 10 ROIs were sampled from 10 independent Poisson processes, whereas during high synchrony states the activation times were sampled from a single Poisson process. As a result, when the 1-bit signal was in the 0 state, the 10 ROIs were activated at independent times, whereas when the 1-bit signal was in the 1 state, the 10 ROIs were activated synchronously. But, importantly, the rate of activation of the ROIs was identical in the two states.

To encode the 1-bit signal using the rate of activation, we used a low rate for the 0 state and a high rate for the 1 state (Fig. 3c). Specifically, we always sampled the activation times of the ROIs from 10 independent Poisson processes, but for the 0 state, we sampled with a 0.5 Hz rate, and for the 1 state we sampled with a 5 Hz rate. This meant that the average rates were approximately the same as for the synchrony encoding (2.7 Hz), but the rates changed depending on the state of the 1-bit signal. Interestingly, we found that, in comparison to whole-field illumination protocols, these patterned optical illumination protocols produced qualitatively more natural responses in the recorded neurons, which was confirmed by analyzing the spectral densities of the whole-cell recordings (Fig. S4). Therefore, using our synchrony and rate encoding protocols, we could investigate the extent to which different subtypes of neurons in layer 2/3 barrel cortex are sensitive to information encoded with the synchrony or rate of activity.

**Responses to synchrony encoding differ between neuron subtypes**. We first examined the responses of FS GAD67+, RS GIN+

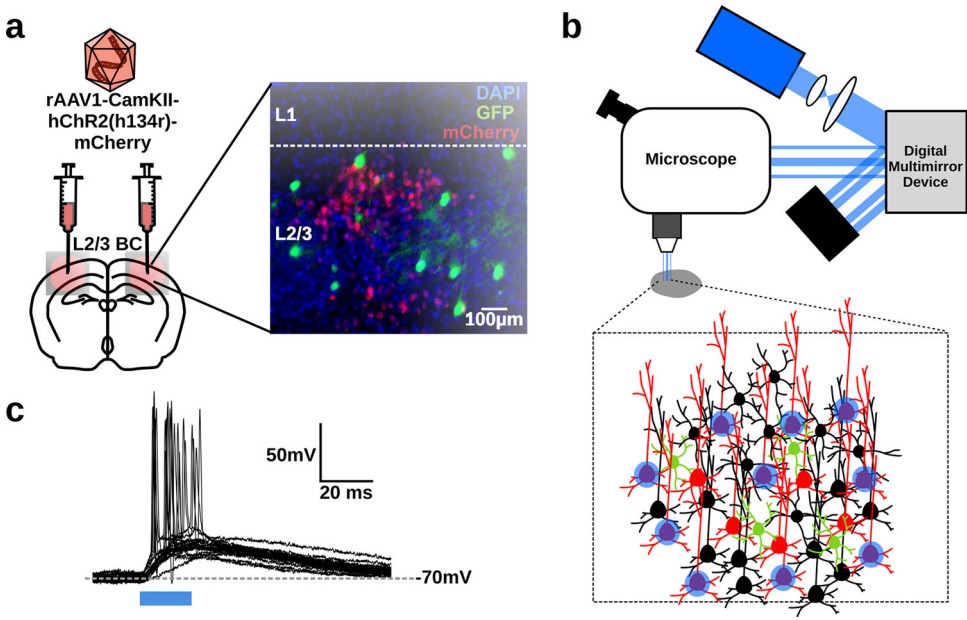

**Fig. 2 Patterned optical stimulation of ChR2+ Layer 2/3 Pyramidal Neurons. a** Viral transfection of Channelrhodopsin-2 into L2/3 Pyramidal Neurons. **b** Illustration of patterned optical stimulation protocol. **c** Sample responses from ChR2+ layer 2/3 pyramidal neurons ($n = 19$) to a 15 μm spot placed directly over the soma of the recorded neuron.

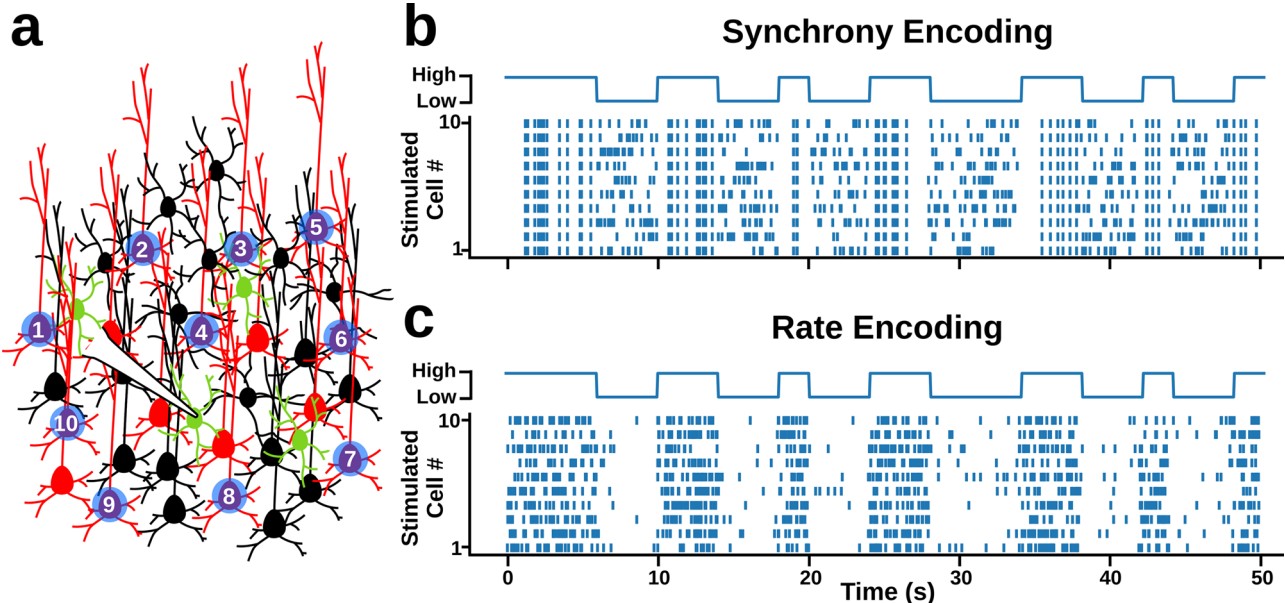

**Fig. 3 Artificially encoding a random 1-bit signal using the synchrony or rate of optical inputs to ChR2+ pyramidal neurons in layer 2/3. a** Experimental protocol. **b** Sample raster plots illustrating how the 1-bit signal was encoded using synchrony of optical inputs. **c** Sample raster plots illustrating how the 1-bit signal was encoded using the rate of optical inputs.

cells, and non-fast-spiking GFP−, non-fluorescent (NF) cells (likely pyramidal neurons) to the synchrony encoding protocol (NF: $n = 17$, FS: $n = 21$, RS: $n = 22$). In general, all three types of neurons exhibited reliable responses to the optical stimulation patterns (Fig. 4a). Histograms of both the mean membrane potential and the firing frequency (i.e. spike counts) for each cell type over the course of the 50 ms window illustrated different responses during the 0 and 1 states of the random signal. As expected, we found that both average membrane potential and spiking frequency were variable and correlated with the number of active ROIs, such that during the 0 state of low synchrony the

spike counts and membrane potential were highly variable, but during the 1 state of high synchrony, the spike counts and membrane potential were less noisy (both were high when all the ROIs were activated, and low when none of the ROIs were activated). More specifically, in the 0 state, the average membrane potential of all three cell types were generally more depolarized and variable (Fig. 4b, light histograms; GFP− = −58.98 mV ± 4.54, GAD67+ = −59.51 mV ± 4.36, GIN+ = −54.96 mV ± 4.92) when compared to neurons in the 1 state (Fig. 4b, dark histograms; GFP− = −65.47 mV ± 2.15, GAD67+ = −66.37 mV ± 2.24, GIN+ = −64.35 mV ± 2.89). Similarly, we found that

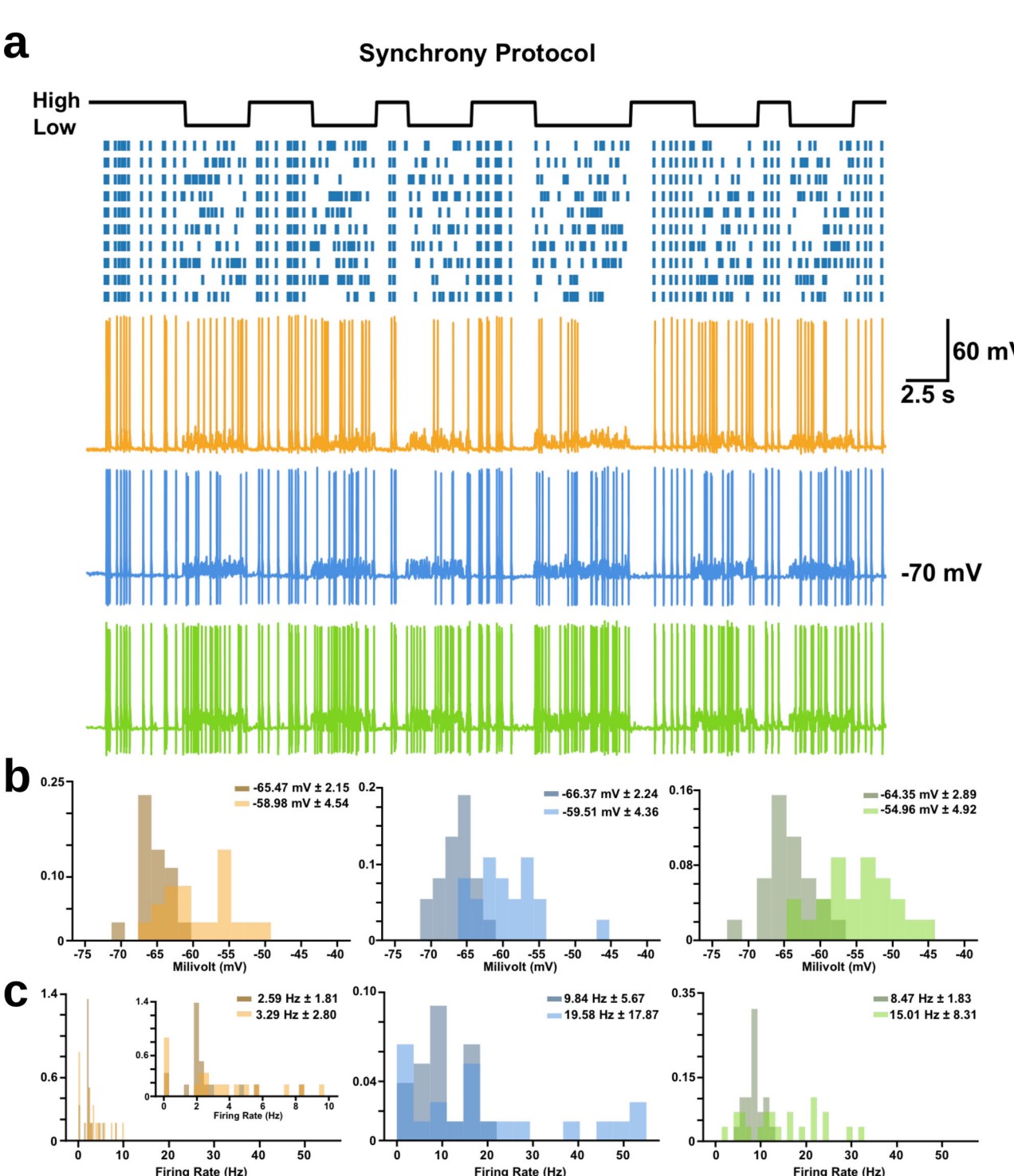

**Fig. 4 High and low states within the synchrony encodings produced different neuronal responses. a** Sample traces of each cell type to the synchrony encoding. **b** Probability density functions of the membrane potential of each cell type during high (darker histogram) and low (lighter histogram) states within the synchrony encodings (numbers shown represent mean membrane potential in each state ± s.d.). Two-way Kolmogorov–Smirnov (2-KS) tests indicate differences in the sample distributions of average membrane potential for each cell type between the 0 and 1 state (GFP−: D(30) = 0.62, $p < 0.001$; GAD67-GFP: D(26) = 0.65, $p < 0.001$; GIN-GFP: D(30) = 0.43, $p = 0.005$). **c** Probability density functions of the firing frequency of each cell type during high and low states within the synchrony encodings (numbers shown represent mean firing frequency in each state ± s.d.). 2-KS tests indicate differences in the sample distributions of firing rates for GIN-GFP cells between the 0 and 1 state, but not GFP− or GAD67-GFP cells (GFP−: D(30) = 0.24, $p = 0.32$; GAD67-GFP: D(26) = 0.31, $p = 0.4$; GIN-GFP: D(30) = 0.43, $p = 0.005$).

during the 0 state of low synchrony, the firing frequency across each cell type was generally higher and extremely variable (Fig. 4c, light histograms; GFP− = 3.29 Hz ± 2.80, GAD67+ = 19.58 Hz ± 17.87, GIN+ = 15.01 Hz ± 8.31), compared to spiking frequencies during the 1 state of high synchrony, which were more consistent (Fig. 4c, dark histograms; GFP− = 2.59 Hz ± 1.81, GAD67+ = 9.84 Hz ± 5.67, GIN+ = 8.47 Hz ± 1.83). At first glance, these results may be counter-intuitive, since they show that the 0 state of low synchrony induced higher average firing rates. But, careful consideration of the low versus high synchrony states shows that during the low synchrony state there are fewer periods where no stimulation occurs, whereas in the high synchrony state stimulation is less frequent, though it is more consistent and strong when it does occur. This leads naturally to higher, though more variable, firing rates during the low synchrony state.

To determine the extent to which each subtype of neuron was sensitive to information encoded in the synchronicity of neural activity, we used information-theoretic tools. Specifically, we examined the mutual information between the 1-bit signal and different aspects of the patched cells' activity. Because our optical protocol did not include any inhibition, the activation of the 10 ROIs induced prolonged polysynaptic activity in the tissue (Fig. S5). To examine differences in the faster and slower integration properties of each cell type, we divided our analyses into early responses found in the first 5 ms of optical activation and late responses throughout the rest of the window, in which we expected there to be more or less polysynaptic activity, respectively.

When we analysed whether the average membrane potential of each cell type conveyed information about the random signal, we found that each neuron type was roughly equal in the amount of mutual information between the 1-bit signal and their mean voltage, for both the early (Fig. 5a; GFP− = 0.3928 ± 0.1294,

GAD67+ = 0.4206 ± 0.1075, GIN+ = 0.4211 ± 0.0827) and late windows (Fig. 5b; GFP− = 0.4562 ± 0.1353, GAD67+ = 0.4917 ± 0.0796, GIN+ = 0.5079 ± 0.0681). This data suggest that although there may be differences in the responses of each neuron type to the synchrony encoded information, the amount of information they carry in their average membrane potential is equivalent.

Next, we examined the spiking responses of the patched neurons. Again, we split our analyses into approximately early and late time windows. Interestingly, unlike the mean voltage responses, we observed clear differences between neuron types in the mutual information between the 1-bit signal and the spike counts. In the early window, GAD67+ cells showed the highest amount of mutual information with the random 1-bit signal (Fig. 5c; GFP− = 0.0218 ± 0.0187, GAD67+ = 0.0861 ± 0.0954, GIN+ = 0.0338 ± 0.0332). These data suggest that when we consider spiking behaviour, GAD67+ interneurons rapidly convey more information than either GIN+ interneurons or pyramidal neurons about signals encoded with synchronous activity.

Meanwhile, for the late time window, the GAD67+ and GIN+ cells showed equal levels of mutual information with the 1-bit signal, both of which were higher than the information contained in the spike counts of the GFP− neurons (Fig. 5d; GFP− = 0.0580 ± 0.0641, GAD67+ = 0.1688 ± 0.1242, GIN+ = 0.2472 ± 0.1049    GFP− = 0.0580 ± 0.0641, GAD67+ = 0.1679 ± 0.1254, GIN+ = 0.2472 ± 0.1049). This suggests that both interneuron subtypes may be more sensitive to signals encoded with synchronous activity than pyramidal neurons. However, it should be noted that in the late time window there may be differences in the rate of synaptic activity thanks to the reverberation of activity in the tissue. This could include not only the recorded neurons or the 10 ROIs themselves but also neurons activated by them. So, whether these differences reflect a different sensitivity to synchronous activity, or different sensitivity to other cells driven

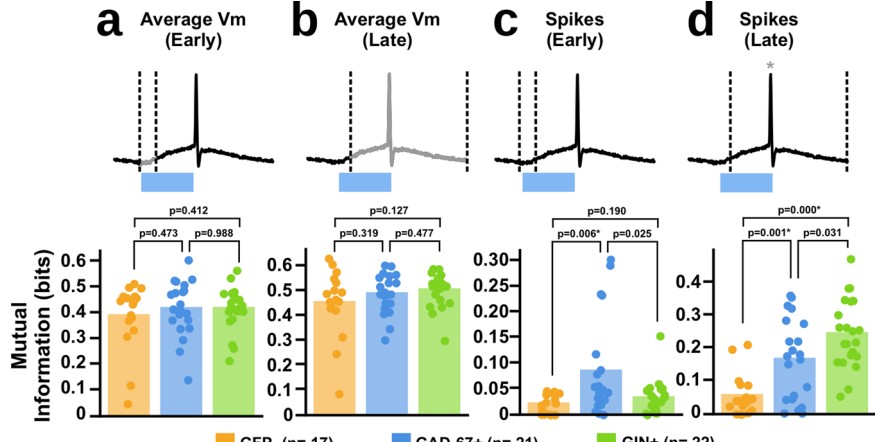

**Fig. 5 Different subtypes of interneurons carry different amounts of information in response to our synchrony encoding. a** Mutual information analysis of the average membrane potential of each cell type to our 1-bit signal within early (0–5 ms) response. Dotted lines indicate window of analysis, with greyed areas indicating what part of the response was analysed (One-way ANOVA, F(2,57) = 0.432, p = 0.656; post-hoc t-tests: GFP− vs. GIN+: t(35) = −0.830, p = 0.412; GAD67+ vs. GIN+: t(41) = −0.00, p = 0.988; GFP− vs. GAD67+: t(36) = −0.737, p = 0.473, * = tests significant at p ≤ 0.017 with Bonferroni correction). **b** Same analysis as (**a**) but restricting the analysis to only the later (5–50 ms) responses (One-way ANOVA, F(2,57) = 1.40, p = 0.244; post hoc t-tests: GFP− vs. GIN+: t(37) = −1.56, p = 0.127; GAD67+ vs. GIN+: t(41) = −0.809, p = 0.477; GFP− vs. GAD67+: t(36) = −0.933, p = 0.319; * = tests significant at p ≤ 0.017 with Bonferroni correction). **c** Mutual information analysis of the spike counts of each cell type to our 1-bit signal within early (0–5 ms) response (Kruskal–Wallis, H(2) = 9.88, p = 0.007; post-hoc t-tests: GFP− vs. GIN+: t(37) = −1.35, p = 0.190; GAD67+ vs. GIN+: t(41) = 2.37, p = 0.025; GFP− vs. GAD67+: t(36) = −3.01, p = 0.006*, * = tests significant at p ≤ 0.017 with Bonferroni correction). **d** Same analysis as (**c**) but restricting the analysis to only the later (5–50 ms) responses (Kruskal–Wallis, H(2) = 21.2, p = 0.001; post-hoc t-tests: GFP− vs. GIN+: t(37) = −6.94, p ≤ 0.001*; GAD67+ vs. GIN+: t(41) = −2.26, p = 0.031; GFP− vs. GAD67+: t(36) = −3.49, p = 0.001*, * = tests significant at p ≤ 0.017 with Bonferroni correction).

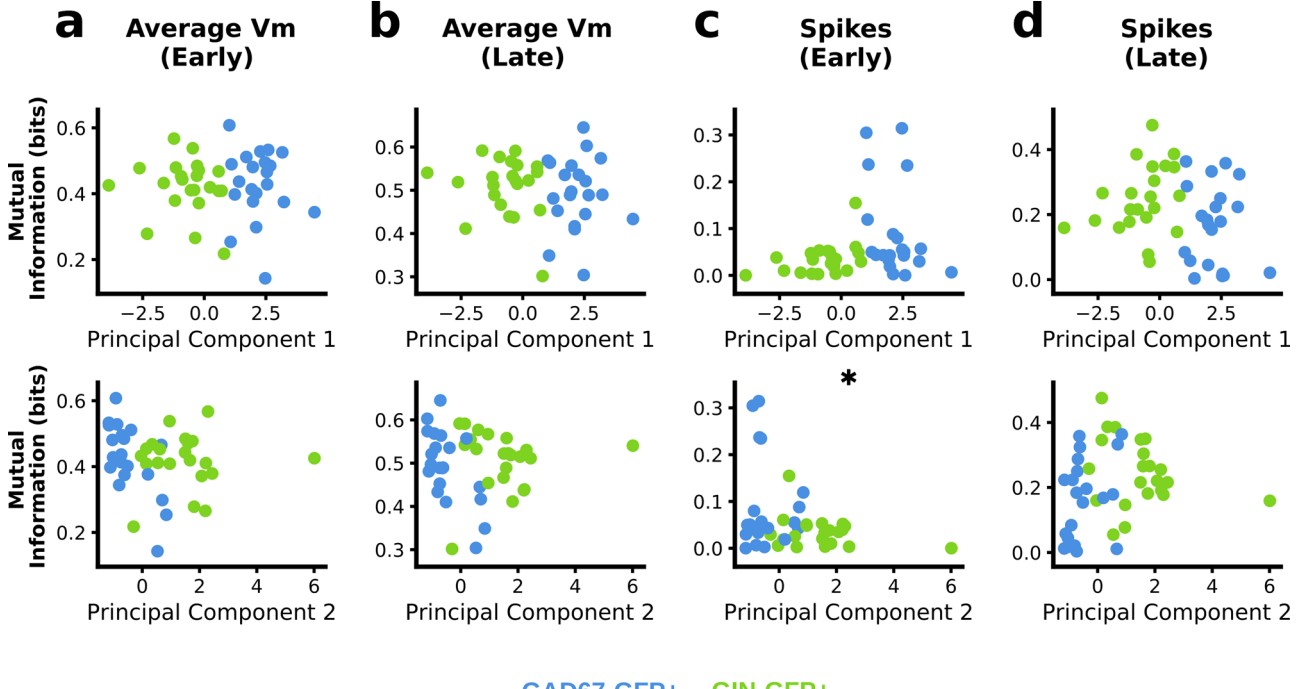

GAD67-GFP+ GIN-GFP+

**Fig. 6 Mutual information between early spike counts in recorded neurons and synchrony code correlates with second principal component of electrophysiological features in GAD67+ and GIN+ interneurons. a** Mutual information between average membrane potential and 1-bit synchrony coded signal correlated with first and second principal components (PC1 and PC2, respectively) of electrophysiological features in the early window (PC1: Pearson's $r = -0.04$, $p = 0.818$; PC2: Pearson's $r = -0.17$; $p = 0.281$). **b** Same as (**a**) but comparing later average membrane potential responses (PC1: Pearson's $r = -0.13$; $p = 0.394$. PC2: Pearson's $r = -0.09$, $p = 0.583$). **c** Conditional mutual information between spike counts and 1-bit synchrony coded signal correlated with first and second principal components of electrophysiological features in the early window (PC1: Pearson's $r = 0.27$, $p = 0.084$; PC2: Pearson's $r = -0.32$, $p = 0.037$). **d** Same as (**c**) but comparing spike counts in later window (PC1: Pearson's $r = -0.21$, $p = 0.169$; PC2: Pearson's $r = 0.20$, $p = 0.191$).

by synchronous activity, is impossible to know. Nonetheless, taken together with the data from the early time window, we can say that our results demonstrate that GAD67+ interneurons, GIN+ interneurons, and GFP- neurons carry different amounts of information over different time-scales about a 1-bit signal that has been encoded via optical activation of 10 ROIs in a synchronous vs. non-synchronous manner.

To give an indication of how the electrophysiological composition of interneurons contribute to their sensitivity to synchrony coding, we assessed how well mutual information estimates correlated with the first and second principal components of electrophysiological features identified for each cell from current injection (Fig. 6). As a reminder, the more positive projections onto the first principal component indicates higher f–I slope, lower membrane time constant, faster spikes, and higher rheobase, whereas more positive projections onto the second principal component indicates higher input resistance, lower cell capacitance, higher sag amplitude and higher adaptation ratio. We found that there was significant negative correlation between the second principal component and mutual information with early window spikes (Fig. 6c, Pearson's $r = -0.32$, $p = 0.037$). This suggests that higher input resistance and stronger presence of slow variables determining the electrophysiological properties of the cell (indicated by sag amplitude and adaptation ratio) weakens sensitivity to synchronous activity over short time scales.

Finally, in order to reinforce these analyses with a more concrete, mechanistic understanding, we constructed a computational model of FS and RS interneurons with electrophysiological properties that matched the neurons in our recordings (Fig. S7a;

see 'Methods'). We then ran simulations with these model cells wherein we provided them with inputs that matched our optical activation patterns (Fig. S7b). Interestingly, we found that the models cells exhibited the same phenomena as the real neurons: when using a synchrony encoding, simulated FS neurons carried more information about the 1-bit signal in a short time-window than simulated RS neurons. Moreover, the models allowed us to actively manipulate the parameters in an 'ablation' study in order to determine which physiological parameters were most important for inducing this difference in information processing. We found that the most important parameters for inducing the FS versus RS mutual information profiles were spike adaptation and spike threshold (Fig. S7c). Given that the first two principal components in our analyses above included these variables, our data suggest that the spiking properties of these two interneuron types can explain the differences we observed. The importance of spiking properties for determining the mutual information results was further supported by additional analyses demonstrating that the amount of mutual information between the 1-bit signal and the spikes of the neurons, but not their average membrane potential, was strongly correlated with the mean firing-rate of the neurons (Fig. S8). Thus, altogether, our data and modelling results suggest that FS GAD67+ are better placed to respond to synchrony codes over short time-scales than RS GIN+ interneurons due to their spiking properties.

**Responses to a rate encoding differ between neuron subtypes.** We then examined responses (GFP−: $n = 17$, GAD67+: $n = 20$, GIN+: $n = 22$) to the same one-bit signal encoded via the rate of

activation of the ROIs. Similar to responses seen in the synchrony optical encoding, each cell type showed different responses during the 0 or 1 state of the rate encoding. However rate-based activation of ROIs induced responses that were of lower magnitude and less noisy during the 0 state (low rate) than the responses during the 1 state (high rate) (Fig. 7a). More specifically, membrane potentials during the 0 state were often hyperpolarized and had low variance (Fig. 7b, light histograms; GFP− = −66.38 mV ± 2.15, GAD67+ = −67.38 mV ± 2.50, GIN+ = −65.81 mV ± 2.34) whereas during the 1 state they were depolarized with high variance (Fig. 7b, dark histograms; GFP− = −54.71 mV ± 5.29, GAD67+ = −56.03 mV ± 6.67, GIN+ = −49.06 mV ± 4.30). This pattern was also seen with the firing rates of each cell type in response to low rates of optical stimulation (Fig. 7c, light histograms; GFP− = 0.90 Hz ± 0.74, GAD67+ = 4.61 Hz ± 4.79, GIN+ = 4.68 Hz ± 2.56) versus high rates (Fig. 7c, dark histograms; GFP− = 6.56 Hz ± 5.20, GAD67+ = 32.65 Hz ± 31.49, GIN+ = 22.57 Hz ± 8.41).

Before conducting information-theoretic analyses, we sought to ensure that any information about the signal encoded by the recorded response was driven by the increase in the rate of optogenetic activation, rather than the unavoidable increase in synchronous stimulation of ROIs with increased rate (see Fig. S6). Specifically, when a rate coding system is used, higher rates will inevitably lead to a larger number of ROIs being activated synchronously. As a result, unlike the synchrony code where one can manipulate synchrony while leaving the rate constant, it is impossible to manipulate the rate while leaving the synchrony constant. As such we conditioned our mutual information measure on the ROI activation count in each time bin (see Materials and Methods). This conditioning was important, because without it the mutual information estimated in the responses to the rate code would have included information that results from synchrony of ROI activation, rather than rate of ROI activation (Figs. S9 and S10).

Conditional mutual information analysis revealed that GAD67+ interneurons tended to have lower conditional mutual information with the 1-bit signal than the other cell types in both the early (Fig. 8a; GFP− = 0.1682 ± 0.0286, GAD67+ = 0.1161 ± 0.0519, GIN+ = 0.1755 ± 0.0278) and late (Fig. 8b; GFP− = 0.0802 ± 0.0238, GAD67+ = 0.0590 ± 0.0395, GIN+ = 0.0782 ± 0.0214) windows. These data indicate that the membrane potential fluctuations of GAD67+ interneurons are less sensitive to signals encoded by rates of activity than GIN+ interneurons and GFP− cells.

We then examined the spiking responses of our recorded neurons, again splitting spike times into those occurring during the early and late windows. Spiking was more frequent in the 1 state than in the 0 state for all neurons, and spike counts tended to increase monotonically with the number of ROIs activated. The purpose of conditioning on ROI activation count is to determine whether there was a difference in spike counts accounted for only by the rate of ROIs activated, and not the number of ROIs activated. We found that for spikes occurring during the early window, there was little difference in conditional mutual information with the one-bit signal between GAD67+ and GIN+ interneurons, although both tended to carry more information about the signal than GFP− cells (Fig. 8c; GFP− = 0.0010 ± 0.0030, GAD67+ = 0.0085 ± 0.0089, GIN+ = 0.0097 ± 0.0172). This indicates that rapid responses to rate encoded pyramidal cell activity are similar between GAD67+ and GIN+ interneurons.

Interestingly, for spikes occurring during the late time window, SST+ interneurons carried more information about the signal than both the GAD67+ and GFP− cells, which carried similar amounts of information (Fig. 8d; GFP− = 0.0091 ± 0.0106,

GAD67+ = 0.0126 ± 0.0102, GIN+ = 0.0463 ± 0.0243). This indicates that GIN+ interneurons can accumulate information about rate encoded signals over longer time windows than both GAD67+ and GFP− cells.

Once again, we also investigated how the electrophysiological composition of interneurons contribute to their sensitivity to rate coding by assessing how well mutual information estimates correlated with the first and second principal components of electrophysiological features (Fig. 9). We found that mutual information between average membrane potential and the rate coded signal was negatively correlated with the first principal component and positively correlated with the second principal component in both the early and late windows (Fig. 9a, b). Furthermore, we also found that mutual information between spike counts in the late window was negatively correlated with the first principal component, and positively correlated with the second principal component. Together, these results indicate that stronger linear filtering properties, along with slower spiking and nonlinear integration, increase the sensitivity to rate-coded signals.

## Discussion

Using a digital micromirror device, we performed ex vivo experiments in slices of mouse barrel cortex to examine the sensitivity of different interneuron subtypes to information encoded with the synchrony or rate of activated neurons. Using GAD67-GFP and GIN-GFP transgenic mice coupled with viral infection of neurons with ChR2, we were able to examine the responses of layer 2/3 fast-spiking GAD67+ and regular-spiking GIN+ interneurons (as well as NF, GFP− cells that were likely pyramidal neurons, Fig. 1). We examined their responses to a 1-bit random signal encoded with either the synchrony or the rate of optical ROI activation (Figs. 2, 3). We found that there were indeed differences between cell types in the amount of information carried about the 1-bit signal. When the signal was encoded using the synchrony of ROI activation, all of the cell types carried similar amounts of information in their membrane potentials, but spiking responses showed differences (Figs. 4, 5). GAD67+ interneurons carried more information than the other cell types during an early time-window, while both GAD67+ and GIN+ interneurons carried more information than GFP− cells in a later time-window. This effect could be driven by the lower input resistance and more linearized integration properties (via absence of slow variables) of FS GAD67+ interneurons (Fig. 6). When the signal was encoded with the rate of ROI activation, we found that GAD67+ interneurons carried less information than either GIN+ or GFP− cells in their membrane potential. For spiking responses, both GAD67+ interneurons and GIN+ interneurons carried more information than GFP− cells in the early window, but in the later time-window, GIN+ interneurons carried more information than either GFP− or GAD67+ cells (Figs. 7, 8). These differences could be driven by the stronger presence of nonlinear slow dynamic variables and longer integration properties of RS GIN+ interneurons (Fig. 9). Altogether, these results demonstrate that there are differences between neocortical cell types in their sensitivity to information encoded with the synchrony or rate of activation.

Our findings are broadly in-line with what is known about neuronal subtypes in the neocortex. First, we found that both interneuron types tended to carry more information than the GFP− cells in their spiking responses. This fits with a previous study that reported higher amounts of information about sensory stimuli in barrel cortex inhibitory interneurons compared to excitatory neurons[35]. (Though it should be noted that that study reported no significant difference in information content between

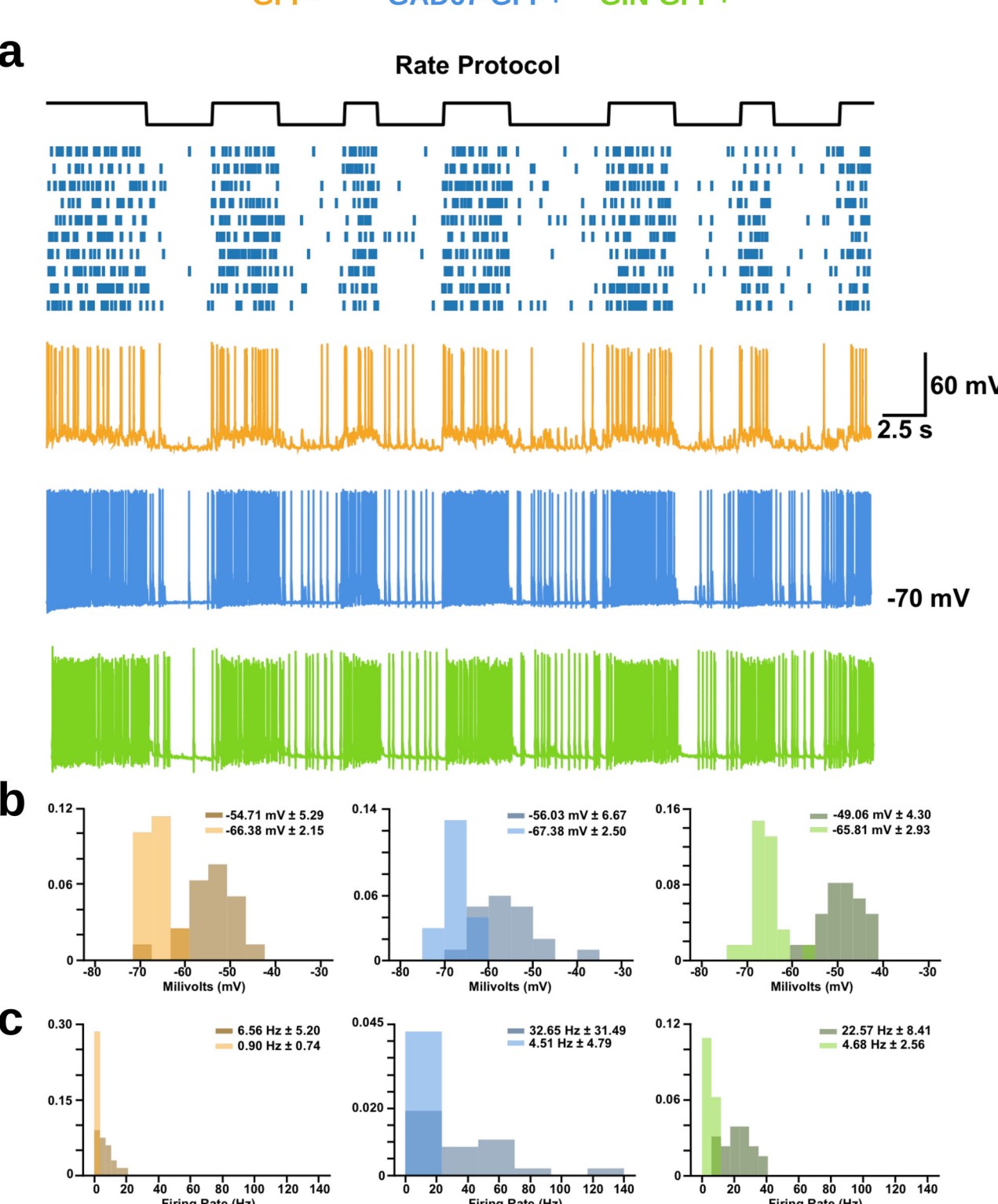

**Fig. 7 High and low states within the rate encodings produced different neuronal responses. a** Sample traces of each cell type to the rate encoding. **b** Probability density functions of the membrane potential of each cell type during high (darker histogram) and low (lighter histogram) states within the rate encodings (numbers shown represent mean membrane potential in each state ± standard deviation). 2-KS tests indicate differences in the sample distributions of average membrane potential for each cell type between the 0 and 1 state (GFP−: D(30) = 0.76, *p* < 0.001; GAD67-GFP: D(26) = 0.80, *p* < 0.001; GIN-GFP: D(30) = 0.83, *p* < 0.001). **c** Probability density functions of the firing frequency of each cell type during high and low states within the rate encodings (numbers shown represent mean firing frequency in each state ± standard deviation). 2-KS tests indicate differences in the sample distributions of average membrane potential for each cell type between the 0 and 1 state (GFP−: D(30) = 0.55, *p* < 0.001; GAD67-GFP: D(26) = 0.52, *p* < 0.001; GIN-GFP: D(30) = 0.73, *p* < 0.001).

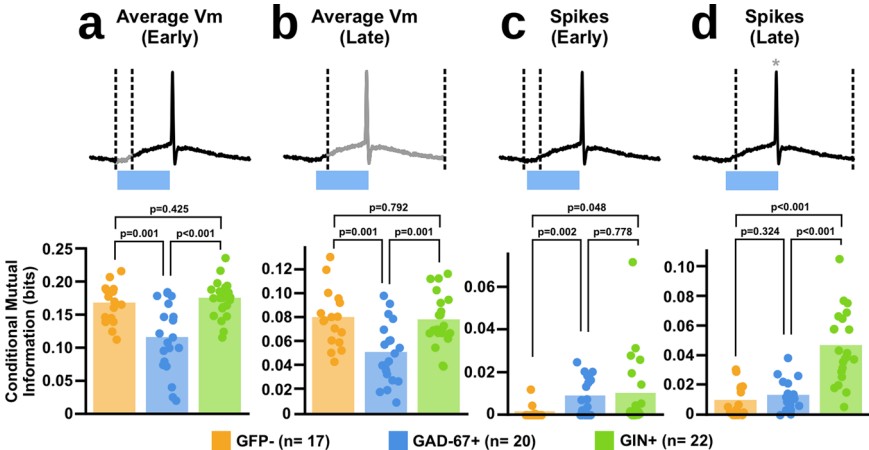

**Fig. 8 Different subtypes of interneurons carry different amounts of information in response to our rate encoding. a** Mutual information analysis of the average membrane potential of each cell type to our 1-bit signal within early (0–5 ms) response. Dotted lines indicate window of analysis, with greyed areas indicating what part of the response was analysed (Kruskal–Wallis, $H(2) = 13.9$, $p \le 0.001$; post-hoc $t$-tests: GFP− vs. GIN+: $t(37) = -0.806$, $p = 0.425$; GAD67+ vs. GIN+: $t(40) = -4.17$, $p \le 0.001^*$; GFP− vs. GAD67+: $t(35) = 3.48$, $p = 0.001^*$, $* =$ tests significant at $p \le 0.017$ with Bonferroni correction). **b** Same analysis as (**a**) but restricting the analysis to only the later (5–50 ms) responses (One-way ANOVA, $F(2,56) = 3.10$, $p = 0.053$; post-hoc $t$-tests: GFP- vs. GIN+: $t(37) = 0.266$, $p = 0.792$; GAD67+ vs. GIN+: $t(40) = -1.98$, $p = 0.054$; GFP− vs. GAD67+: $t(35) = 1.93$, $p = 0.062$, $* =$ tests significant at $p \le 0.017$ with Bonferroni correction). **c** Mutual information analysis of the spike counts of each cell type to our 1-bit signal within early (0–5 ms) response (One-way ANOVA, $F(2,56) = 2.93$, $p = 0.066$; post-hoc $t$-tests: GFP− vs. GIN+: $t(37) = -2.04$, $p = 0.048$; GAD67+ vs. GIN+: $t(40) = -0.226$, $p = 0.778$; GFP− vs. GAD67+: $t(35) = -3.66$, $p = 0.002^*$, $* =$ tests significant at $p \le 0.017$ with Bonferroni correction). **d** Same analysis as (**c**) but restricting the analysis to only the later (5–50 ms) responses (Kruskal–Wallis, $H(2) = 30.5$, $p \le 0.001$; post-hoc $t$-tests: GFP− vs. GIN+: $t(37) = -6.42$, $p \le 0.001^*$; GAD67+ vs. GIN+: $t(40) = -5.92$, $p \le 0.001^*$; GFP− vs. GAD67+: $t(35) = -1.14$, $p = 0.324$, $* =$ tests significant at $p \le 0.017$ with Bonferroni correction).

interneurons and excitatory neurons in layer 2/3, which we have studied here.) Second, our finding that GAD67+ interneurons' spikes rapidly convey information about a signal encoded with synchronous activity, while GIN+ interneurons gradually accumulate information about a signal encoded with different rates of activity, fits with what is broadly known about the biophysics of these cell types. Specifically, the rapid membrane time-constants, and rapid spiking properties of FS PV+ interneurons[36] fits with rapid transmission of information about synchronous inputs, while the adaptive spiking responses of SST+ interneurons[24] fits with long latency responses to high rate inputs[37,38]. Moreover, our computational model (Fig. S7) and analyses of the relationship between firing-rate and mutual information (Fig. S8), further support the idea that differences in spiking properties can explain the differences we observed between FS and RS cell types.

Our data suggest that there are potential divisions of labour in information encoding in the neocortical microcircuit. Both GAD67+ and GIN+ subtypes carried information about the 1-bit signal regardless of the encoding format we used, but our results imply that, roughly, GAD67+ cells, which are FS interneurons, rapidly provide more information about synchronous network activity, while GIN+ interneurons gradually accumulate information about the rate of activity in the surrounding neurons. Given this, the synchronous activation of excitatory neurons in the circuit[39,40] may activate perisomatic inhibition more strongly, while recurrent or top-down signals that build-up over time may activate distal dendritic inhibition more strongly. This would also suggest that reports that synchrony and rate can carry distinct information about sensory stimuli[9,12] may link certain aspects of sensory stimuli with certain forms of inhibition to pyramidal neurons. Future work should examine whether the information carried by FS and RS interneurons reflects signals communicated by the synchrony and rate of pyramidal cell activation, respectively. Such investigations would ideally be done using similar patterned optical activation approaches to ours, but in vivo, as data suggests that patterned optical activation can reveal aspects of

network organization and function that other forms of stimulation cannot, particularly with respect to inhibitory microcircuits[41,42].

One aspect of interneuron specialization for information coding that we have not explored in this paper is the role of network-level interactions between interneuron classes. For example, it is known that PV+ and SST+ interneurons can be connected to one another[28], which would mean that the coding properties of one neuron type could affect the other directly. This may have been the case in our data here, given that we could not guarantee that polysynaptic interactions were not involved in generating some of the information processing we observed. However, our computational model (Fig. S7) suggests that at least the core aspects of the information coding specialization observed here can be explained in a cell-autonomous manner. Nonetheless, network interactions are an important consideration here, especially when we consider the potential role of oscillations in shaping synchrony and rate across a network, and the role of interneurons in regulating oscillations[43].

It is important to note that our study was limited by a number of factors. First, we were performing our experiments ex vivo, and there are likely important differences in interneuron activity in vivo that relate to factors such as movement or neuromodulation[23]. Second, because we were using single-photon excitation with no optical inhibition, we could not guarantee fine-grained optogenetic control over a small population of excitatory neurons (Fig. S2), so it must be recognized that we were likely activating the recorded neurons directly sometimes, and certainly we were activating more than 10 presynaptic neurons and triggering recurrent, polysynaptic activity. It should, however, be noted that any lack of control in the specific spike-timing and population size of activated neurons does not affect our findings of different sensitivity to the encoded information, since all tissue samples used in this study are affected by these sources of variability equally. Third, our dataset did not include morphological information for the neurons, which also helps to determine classifications of interneurons[17]. As such, the cells in our study may have actually belonged to subclasses which

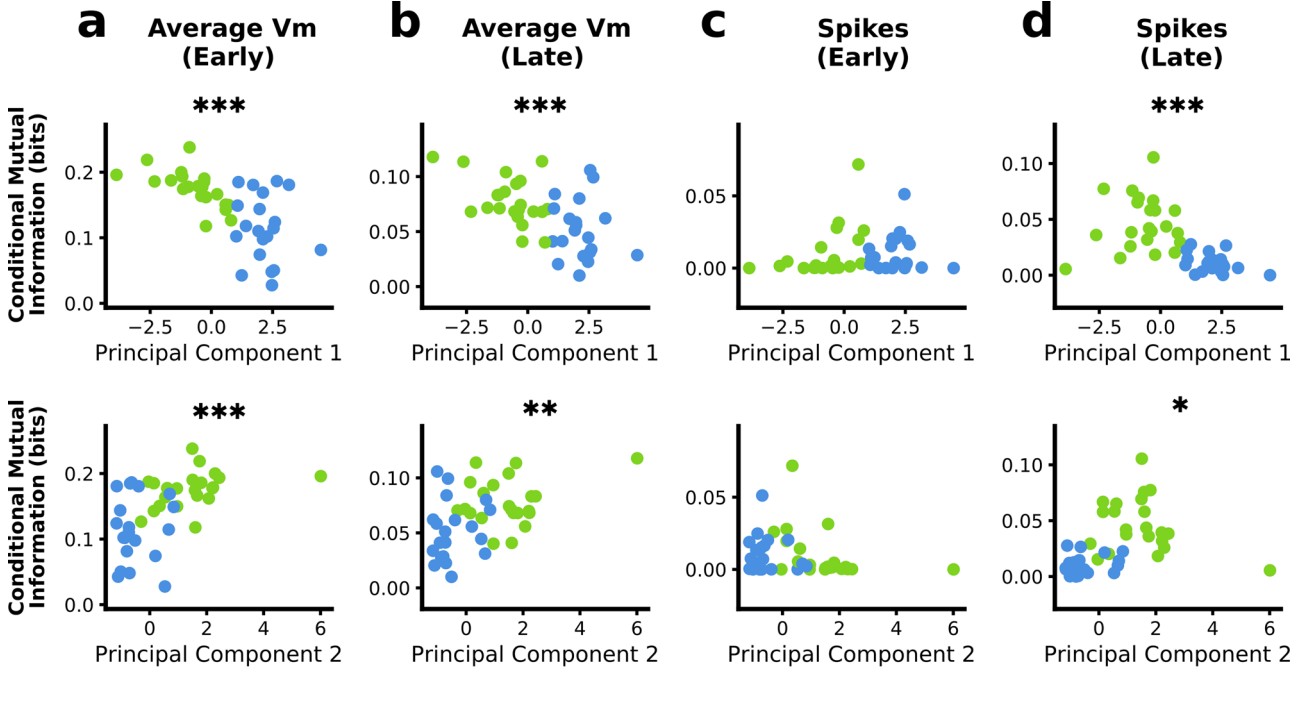

**Fig. 9 Conditional mutual information between responses in recorded neurons and rate code correlates with first and second components of electrophysiological features in GAD67+ and GIN+ interneurons. a** Conditional mutual information between average membrane potential and 1-bit rate coded signal correlated with first and second principal components (PC1 and PC2, respectively) of electrophysiological features in the early window (PC1: Pearson's $r = -0.65$, $p \leq 0.001$; PC2: Pearson's $r = 0.52$; $p \leq 0.001$). **b** Same as (**a**) but comparing later average membrane potential responses (PC1: Pearson's $r = -0.54$; $p \leq 0.001$. PC2: Pearson's $r = 0.46$, $p = 0.002$). **c** Conditional mutual information between spike counts and 1-bit rate coded signal correlated with first and second principal components of electrophysiological features in the early window (PC1: Pearson's $r = 0.19$, $p = 0.237$; PC2: Pearson's $r = -0.28$, $p = 0.078$). **d** Same as (**c**) but comparing spike counts in later window (PC1: Pearson's $r = -0.56$, $p \leq 0.001$; PC2: Pearson's $r = 0.36$, $p = 0.018$).

could have further refined our results. Hopefully, the relationship between morphology and information coding specialization will be explored by follow-up studies. And finally, our findings cannot directly inform us about whether these codes are actually used for computation in vivo. That requires behavioural responses from animals to determine whether downstream circuits utilize the information encoded with rate or synchrony[44]. Nonetheless, we believe that our findings are informative and can help to guide future work that attempts to examine potential divisions in coding in the neocortical circuit in vivo.

In summary, we found evidence that different subtypes of neurons in the neocortical microcircuit are differentially sensitive to information encoded with the synchrony or rate of activity in the surrounding network. This supports the idea that the brain is capable of multiplexing information through activation of distinct interneuron subtypes.

## Methods

**Animals**. GIN-GFP (FVB-Tg(GadGFP)45704Swn/J; JAX#003718) and GAD67-GFP animals (CB6-Tg(Gad1-EGFP)G42Zjh/J; JAX#007677) were obtained from Jackson Laboratory. Mice were weaned at 21 days, in a temperature controlled room with a 12 h light/dark cycle. Mice were given food and water, ad libitum. All procedures were in accordance with the regulations provided by the Canadian Council for Animal Care and approved by the Local Animal Care Committee at the University of Toronto Scarborough.

**Viral Microinfusion**. Mice 5–7 weeks old received bilateral microinfusion of AAV1-CamKii-hChR2(H134R)- mCherry.WPRE.hGH (Addgene, #26975-AAV1) into layer 2/3 of the barrel cortex (−1.3 mm AP, ±3.1 mm ML, −1.1 mm DV).

Mice were treated with ketoprofen (5 mg/kg) and anesthetized with isofluorane (4% induction, 2% maintenance). The anesthetized animal was then placed on a stereotaxic frame (Stoelting) and holes drilled in the skull above the coordinates of interest. To inject the viral vectors, a Hamilton Neuros Syringe (Hamilton, #65460-05) was connected to a microinjector (QSI, Stoelting) to infuse the virus at a volume of 0.15 µL per side with a rate of 0.05 µL/min. After each injection, the syringe was left in the brain for another 5 min to allow for sufficient diffusion of the virus. Following surgery, mice were treated with 0.5 ml of 0.9% saline sub-cutaneously and received ketoprofen post-operatively for 2–3 days. Based on the extent of mCherry expression, we typically observed expression up to 200–250 µm away from the injection site. This encompassed layer 2/3, but also layer 4, and in a small number of animals, the superficial portion of layer 5.

**Immunohistochemistry**. To confirm whether GIN-GFP and GAD67-GFP targeted SST+ and PV+ cells, respectively, brains from each transgenic line were fixed with 4% paraformaldehyde (PFA) via transcardial perfusion. After 2 days of fixation, the brains were sliced at 50 µm thickness using a vibratome (Leica).

For PV staining, free-floating brain sections of the barrel cortex were first washed in PBS and then incubated in 1% $H_2O_2$ in PBS for 30 min at room temperature. Slices were then blocked with PBS containing 10% goat serum, 3% bovine serum albumin, and 0.05% Triton-X-100 for 2 h at room temperature. Afterwards, sections were incubated in PBS blocking buffer containing mouse anti-PV primary antibody (Thermofisher, 1:500) overnight at 4 °C. The next day, slices were washed in PBS and then incubated in PBS blocking buffer containing goat anti-rabbit secondary antibody conjugated with an Alexafluor 594 (Life Technologies, 1:500) for 1 h at room temperature.

For SST staining, free-floating brain sections of the barrel cortex were first washed in PBS and then incubated in 1% $H_2O_2$ in PBS for 30 min at room temperature. Slices were then blocked with the same blocking solution as above for 2 h at room temperature. Afterwards, sections were incubated in PBS blocking buffer containing mouse anti-SST primary antibody (Novus, 1:500) overnight at 4 °C. The next day, slices were washed in PBS and then incubated in PBS blocking buffer containing goat anti-rabbit secondary antibody conjugated with an

Alexaflour 594 (Life Technologies, 1:500) for 1 h at room temperature. Following this, tyramide signal amplification (TSA) was performed by incubating the sections in Rhodamine TSA reagent (1:30,000, diluted in 0.1 M Borate Buffer with 0.01% $H_2O_2$) for 30 min at room temperature.

Following staining, slices were washed with PBS, mounted onto gelatin-coated slides, and covered with a coverslip using Fluoshield (Sigma-Aldrich). Images were obtained using a confocal laser scanning microscope (Zeiss) with a ×10 objective.

For cell counting experiments, L2/3 of the barrel cortex was imaged and was counted for GFP+, PV+ and SST+ cells. Approximately 4–6 sections/mouse were counted and averaged, with 4–6 mice/group. Genotypic specificity (total numbers of PV+ or SST+ cells/total numbers of GFP+ cells × 100), and efficiency (total numbers of GFP+ cells/total numbers of PV+ or SST+ cells × 100) were calculated using these averages.

**Ex vivo slice electrophysiology**. Mice (GAD67-GFP: $n = 25$, GIN-GFP: $n = 19$) aged 7–12 weeks were anesthetized with 1.25% tribromoethanol (Avertin) and underwent cardiac perfusion using a chilled cutting solution containing (in mM): 60 sucrose, 25 NaHCO$_3$, 1.25 NaH$_2$PO$_4$, 2.5 KCl, 0.5 CaCl$_2$, 2 MgCl$_2$, 20 D-glucose, 3 Na-pyruvate and 1 ascorbic acid, injected at a rate of ~1 mL/min. After 5–8 min of perfusion, the brain was quickly removed and cut coronally (350 μm thickness) with a vibratome (VT1000S) in chilled cutting solution to obtain slices of the barrel cortex. Once cut, these slices were then transferred into a recovery chamber comprising of a 50:50 mix of warm (34 °C) cutting solution and artificial cerebrospinal fluid (aCSF) containing (in mM): 125 NaCl, 25 NaHCO$_3$, 1.25 NaH$_2$PO$_4$, 2.5 KCl, 1.3 CaCl$_2$, 1 MgCl$_2$, 20 D-glucose, 3 Na-pyruvate, and 1 ascorbic acid. Following 30 min–1 h of incubation, the slices were then transferred into an incubation chamber with room temperature aCSF. Within the recording chamber, aCSF was heated to 32 °C using an in-line heater. Whole-cell current-clamp recordings were made using glass pipettes filled with (in mM): 126 K D-Gluconate, 5 KCl, 10 HEPES, 4 MgATP, 0.3 NaGTP, 10 Na-phosphocreatine. Glass capillary pipettes were pulled with a Flaming/Brown pipette puller with tip resistances between 4 and 8 MΩ. Patched cells were monitored for series resistance throughout recordings, and those that exceeded 30 MΩ were not included in this study.

**Patterned illumination with a digital micromirror device**. To optically encode a 1-bit random signal into the activity of L2/3 neurons, we used one-photon patterned illumination with a digital micromirror device (Polygon400, Mightex). After viral infusion surgery, and 2–3 weeks for expression (see above) we prepared slices and excited ROIs containing mCherry+ neurons in the slice. To target optical activation to as few neurons as possible, while maintaining reliable spiking responses in activated cells, we used software to draw circular ROIs of 15 μm in diameter (PolyScan V2, Mightex) around mCherry+ cells. These ROIs defined discs of illumination. We always drew the ROIs such that only a single mCherry+ neuron could be seen within the ROI. We cannot guarantee that no other cells that expressed ChR2 were ever contained within the ROIs, but we can say that we never observed more than one mCherry+ cell per ROI. The light used to activate the cells was generated from an LED with a 470 nm wavelength, and the optical power at the microscope stage was set to ~14 mW/mm$^2$, which we found to be sufficient for driving reliable spiking responses in targeted neurons.

To test the spatial specificity of our patterned illumination setup and determine whether it could reliably induce spiking, we conducted whole-cell patch-clamp recordings from ChR2+ layer 2/3 pyramidal cells in the barrel cortex while illuminating circular patterns of blue light placed 25 μm apart in sequential order (see Fig. S2) in both the dorsal-ventral axis as well as the medial-lateral axis of the slice (470 nm, ~14 mW/mm$^2$). To determine the probability of spiking based upon the spatial distance of the light spot, the median probability of a spike occurring was calculated as well as the 95% confidence interval for the median via bootstrapping ($n = 1000$).

As such, to limit the spread of optical activation, we chose to limit our recordings to regions of tissue that showed sparse infection. More specifically, we drew ROIs around mCherry+ cells that were >50 μm away from their nearest neighbouring mCherry+ cell, as indicated when using PolyScan V2 software. This helped to ensure the fidelity of spiking soon after the onset of illumination of mCherry+ cells (Fig. S2c). Also, note from Fig. S2c that cells >50 μm away from the ROI would be expected to spike much later than cells centred on the ROI.

**Encoding a 1-bit random signal with synchrony or rate of ROI activation**. To examine the responses of different neuron types to synchrony and rate of optically induced activity we developed protocols for encoding a random 1-bit signal (0 versus 1) in a brain slice. To do this, we drew 10 discs of illumination centred on mCherry+ neurons, that were in close proximity to the mCherry− patched cell. As previously stated, in order to mitigate unintended cross-stimulation of ROIs, we tried to space out the spots from each other by at least 50 μm, as we had previously observed that the probability of spiking dropped significantly if a spot was placed at least 50 μm distance from a cell (Fig. S2). A one-bit signal, s(t), was encoded in the optogenetically-driven activity over 10 ROIs using either a rate or temporal code. Under a rate code, neurons were driven by pulses $x^R_{1:10}(t)$ sampled from 10

independent inhomogenous Poisson process with rates, $\lambda^R(t)$, depending on the value of s(t), such that:

$$\lambda^R(t) = \begin{cases} 5 \text{ Hz}, & \text{if } s(t) = 1 \\ 0.5 \text{ Hz} & \text{if } s(t) = 0 \end{cases} \quad (1)$$

$$x^R(t) \sim \text{Poisson}(\lambda^R(t)) \quad (2)$$

Under a synchrony code, neurons were driven by pulses $x^T_{1:10}$ sampled from 10 independent homogeneous Poisson processes or from a single homogeneous Poisson process, creating states of uncorrelated (s(t) = 0) and perfectly correlated (s(t) = 1) pulses:

$$x^T_{1:10}(t) = \begin{cases} \text{Poisson}(\lambda^T), & \text{if } s(t) = 1 \\ \text{Poisson}(\lambda^T_{1:N}) & \text{if } s(t) = 0 \end{cases} \quad (3)$$

The rate $\lambda^T$ was set to the mean rate of the the rate-coded signal, i.e. $\lambda^T = \mathbf{E}[\lambda^R(t)] \approx 2.7$ Hz.

**Electrophysiological characterization**. Electrophysiological characteristics for each neuron were estimated from 500 ms current injection steps ($I_{inj}$) ranging from −80 pA to 400 pA in 40 pA increments. Eleven features were extracted in total including: (1) resting membrane potential ($V_{rest}$, mV), (2) input resistance ($R_{in}$, MΩ), (3) cell capacitance ($C_{mem}$, pF), (4) membrane time-constant ($\tau_{mem}$, ms), (5) rheobase ($I_\theta$, nA), (6) f–I slope (f′, Hz/nA), (7) spike adaptation ratio, (8) sag amplitude ($V_{sag}$, mV), (9) spike threshold ($V_\theta$, mV), (10) spike amplitude ($V_{amp}$, mV), and (11) spike half-width ($T_{half}$, ms). Standard calculations were used for these features[18,45,46], but briefly, we will note the following for clarity:

- Spike times were identified as times at which membrane potentials crossed −20 mV with a positive gradient.
- Rheobase $I_\theta$ (current at which non-zero spike counts occur) and f–I slope, f′, were estimated by fitting the piecewise-linear scalar function $f(I) = \max(0, f' \cdot (I - I_\theta)$ to spike counts at each $I_i nj$ step value. Many cell types display spike accommodation with non-linear above-rheobase f–I relationships. We defined f–I slope to mean the initial slope above rheobase. Therefore, we fit this function to sub-rheobase and up to the first five above-rheobase spike counts inclusively.
- Spike adaptation ratio was estimated as the ratio between the last and first spike-time intervals (the difference between spike times). This requires a minimum of three spikes to estimate. For spike-trains with ≥7 spikes, the last two and first two intervals were used to estimate the ratio to improve estimate quality. Only the spike train from the highest $I_{inj}$ was used to estimate this feature.
- $V_{rest}$ was estimated as the average membrane potential in the 10 ms prior to current injection.
- $V_{sag}$ was estimated as $|\min(V) - V_{rest}|$ at $I_{inj} = −80$ nA during current injection.
- $V_\theta$ was estimated from all extracted spikes. A window around each identified spike time was used to extract action potential V(t) traces and the z-scored slope $z(V'(t))$ of each action potential was calculated. $V_\theta$ was estimated as the membrane potential at which $z(V'(t)) \geq 0.5$
- $V_{amp}$ was estimated as $V_{amp} = max(V) - V_\theta$
- $T_{half}$ was defined as the duration of an action potential for which $V(t) \geq (V_{amp} - V_\theta)$ and was averaged across all extracted action potentials.
- $R_{in}$ measurements were calculated and averaged across membrane potentials resulting from sub-rheobase, non-zero current injection. $R_{in}$ was calculated using Ohm's law as $(V_\infty - V_{rest})/I_{inj}$ where the steady-state membrane potential $V_\infty$ was estimated as the temporally averaged membrane potential over the last 10 ms of current injection.
- $\tau_{mem}$ measurements were taken from membrane potential decay 100 ms after sub-rheobase non-zero current injection. Estimates were calculated by fitting a single-order exponential function of the form $V(t) = (V_0 - u)\exp(-t/\tau_{mem}) + u$ and averaged
- $C_{mem}$ was calculated as $\tau_{mem}/R_{in}$

.

**Principal components analysis and hierarchical clustering**. Principal components analysis of each cell's electrophysiological characteristics was conducted using the sklearn python package[47]. To generate the dendrogram, we used the scipy.cluster package to implement hierarchical clustering[48]. Ward's method was used to calculate the distance between each cluster.

**Mutual Information**. Mutual information $I(r; s)$[49] between the response r(t) of the recorded neuron and the one-bit signal s(t) was used to assess the sensitivity of the recorded neurons to the synchrony encoding of the signal:

$$I(r; s) = H(r) + H(s) - H(r, s) \quad (4)$$

where H(r), H(s) and H(r, s) are the entropy of the response, signal and joint entropy of the signal and response, respectively. The responses of the neurons, r(t),

were defined as mean membrane potential (e.g. Fig. 7a, b) or the spike counts (e.g. Fig. 7c, d) over the given temporal window.

Since temporal correlations increase amongst inputs when rates increase (Fig. S6), we could not directly calculate the mutual information between the rate of activations and the responses. Instead, the mutual information was conditioned on the number of ROIs activated $y(t) = \sum_{i=1}^{10} x_i(t)$ to discern how sensitive the recorded neuron was to the rate-coded signal:

$$I(r;s|y) = H(r,y) + H(s,y) - H(r,s,y) - H(y) \qquad (5)$$

where $H(r,y)$, $H(s,y)$, and $H(r,s,y)$ are the joint entropy of the response and number of active ROIs, the signal state and number of active ROIs, and the response, signal state, and active ROIs, respectively, whereas $H(y)$ is the entropy of the number of active ROIs.

To estimate the entropy of discrete variables U (such as signal state, number of active ROIs, spike counts), we used the standard definition of entropy as the weighted average of the log probability mass function[49,50], i.e.

$$H(U) = -\sum_u P_U(u)\log_2 P_U(u) \qquad (6)$$

with the convention $P_U(u)\log_2 P_U(u) = 0$ if $P_U(u) = 0$ applied.

Similarly, the entropy of continuous random variables (average membrane potential) was estimated by constructing histograms to approximate the probability density function of the variable with a discrete probability mass function, i.e.:

$$H(u) = -\Delta u \sum_{i=1}^{B} P(u_i - \Delta u/2 \le u < u_i + \Delta u) \log_2 P(u_i - \Delta u \le u < u_i + \Delta u/2) \qquad (7)$$

along histogram bin midpoints $u_i$ with bin width $\Delta u$. The number of bins $B$ was chosen as the maximum of Sturges' formula[51] and the Freedman-Diaconis rule[52]:

$$B = \max(2\frac{IQR(u)}{n^{1/3}}, \log_2 n + 1). \qquad (8)$$

where $n$ is the size of data $u$, and $IQR$ is the interquartile range.

**Statistics and reproducibility**. All data analysis code was written in python 2.7 using tools from the scientific computing ecosystem (numpy[53,54], scipy[48], matplotlib[55], scikit-learn[47], pandas[56], neo[57]).

To determine whether the mean mutual information was different between each cell type, we ran a One-Way ANOVA, or Kruskal–Wallis test if Levene's test indicated unequal variances between groups. We also applied post-hoc individual, two-tailed t-tests for differences between pairs of groups, or Welch's t-test if Levene's test indicated unequal variance. Bonferroni corrections for multiple comparisons were applied for each test. All tests were run using scipy.

**Spiking neuron simulation**. Simulations of spiking neuron were used to examine the effect of intrinsic properties controlling the responsiveness of neurons to rate or synchrony codes. We used adaptive exponential-integrate-and-fire (AdExp)[58–60] neuron models in our simulations since they are sufficiently flexible to reproduce firing statistics of fast-spiking and regular-spiking interneurons, while retaining a low-dimensional parameter space that is simple to tune.

The AdExp Neuron model describes the evolution of the membrane potential $V_m$ and adaptation current $w$ of a point neuron, i.e.,

$$C_m \frac{dV_m}{dt} = -g_L(V_m - E_L) + g_L \Delta_T \exp\left(\frac{V_m - V_T}{\Delta_T}\right) - g_{GLUT}(V_m - E_{GLUT}) - w + I_{inj} \qquad (9)$$

$$\tau_w \frac{dw}{dt} = a(V_m - E_L) - w \qquad (10)$$

where $C_m$ is membrane capacitance, $g_L$ is leak conductance, $E_L$ is leak reversal potential, $V_T$ is a soft threshold membrane potential, $\Delta_T$ is a sharpness parameter controlling exponential activation, $g_{GLUT}$ is the total conductance of excitatory input synapses, $E_{GLUT}$ is the reversal potential of excitatory input synapses, $I_{inj}$ is injected current, $\tau_w$ is the time-constant of adaptation current decay, and $a$ is a parameter coupling the adaptation current to the membrane potential.

As with other integrate-and-fire neuron models, the variables are permitted to evolve under these dynamics until a spike event is triggered, i.e., when $V_m > V_{cut}$, where $V_{cut} = V_T + 5\Delta_T$ to prevent numerical overflow. When this occurs, the membrane potential $V_m$ and adaptation current $w$ are reset according to

$$V_m \leftarrow V_{reset} \qquad (11)$$

$$w \leftarrow w + b \qquad (12)$$

where $V_{reset}$ is the post-spike reset membrane potential, and $b$ is the jump of spike-triggered adaptation.

We used a mixture of parameter estimates from our own data, hand-tuning according to phase portraits in ref. [60], and literature review[61,62] to determine parameters for FS and RS interneurons

We can group subsets of these parameters to describe how they influence neuron behaviour: i) linear leaky integration ($C_m$, $g_L$, $E_L$), ii) spike threshold ($\Delta_T$, $V_T$), iii) adaptation ($\tau_w$, $a$, $b$), and iv) reset ($V_{reset}$). Additionally, we

**Table 1 Table of parameters used to model fast-spiking (FS) and regular-spiking (RS) interneurons using adaptive exponential-integrate-and-fire neurons.**

| Group | Parameter | Neuron Type | |
| | | FS | RS |
|---|---|---|---|
| Leaky integrator | $\bar{C}_m$ | 60 pF | 60 pF |
| | $g_L$ | 6 nS | 3 nS |
| | $E_L$ | −70 mV | −70 mV |
| Threshold | $\Delta_T$ | 1 mV | 4 mV |
| | $\bar{V}_T$ | −43 mV | −62 mV |
| | $\alpha$ | 1/64 | 1/100 |
| Adaptation | $\tau_w$ | 1 ms | 100 ms |
| | $a$ | 0 nS | 0.001 nS |
| | $\bar{b}$ | 0 pA | 40 pA |
| Reset | $V_{reset}$ | −70 mV | −70 mV |

Parameters are split into groups that describe their contribution to the firing statistics of the cell.

incorporated a degree of heterogeneity in neurons by sampling $V_T$, $b$, and $C_m$ from Gaussian distributions $\mathcal{N}(\bar{V}_T, \alpha \bar{V}_T)$, $\mathcal{N}(\bar{b}, \bar{b}/400)$, $\mathcal{N}(\bar{C}_m, \bar{C}_m/100)$ respectively. Parameters for each neuron type are listed in Table 1. These parameters imbue FS neurons with no adaptation, as such dynamics are solely determined by leaky integration and spike-threshold parameters. In contrast, these parameters imbue RS neurons with adapting spike patterns, along with (on average) a larger membrane time constant and lower spike-threshold than FS cells. This corresponds to our own data, in which RS cells have lower rheobase and larger membrane time constants than FS cells.

For simulation of our rate and synchrony coding experiments, we incorporated a population of 10 presynaptic single-exponential excitatory synapses

$$\frac{dg_e^{(i)}}{dt} = -g_e^{(i)}/\tau_e \qquad (13)$$

$$g_e^{(i)} \leftarrow g_e^{(i)} + h \quad (post-spike) \qquad (14)$$

where $g_e^{(i)}$ corresponds to the conductance of the synapse at input $i$, $\tau_e = 5$ ms is the synaptic decay time-constant, and $h$ controls the jump in synaptic conductance post-spike.

Simulations were conducted using Brian 2.4[63], using Euler integration with a time step of 0.1 ms (exact solutions used for synapses).

**Reporting summary**. Further information on research design is available in the Nature Research Reporting Summary linked to this article.

## Data availability
Data for this work is publically available on Dryad (https://datadryad.org/stash/share/OVVBYwXRW7iWnNuxzRsMghIpA2D2QJXUuY59ioNZH04).

## Code availability
Code for this work is publically available on GitHub (https://github.com/lyprince/nc_paper).

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

## Acknowledgements

This work was supported by a Human Frontier Science Program Young Investigator Grant to M.M.K, J.K. and B.A.R. (RGY0073/2015) and a Natural Sciences and Engineering Research Council of Canada Discovery Grant to B.A.R. (RGPIN-2014-04947).

## Author contributions

B.A.R., M.M.K. and J.K. conceived of the experiments and analysis. M.M.T., D.G., L.S. and H.C. collected the data. M.M.T. and L.Y.P. analysed the data. L.Y.P. designed and ran simulations. M.M.T., L.Y.P. and B.A.R. wrote the manuscript. M.M.T., L.Y.P., M.M.K., J.K. and B.A.R. edited the manuscript.

## Competing interests

The authors declare no competing interests.
