## [Peer Review File · Communications Biology]

Reviewers' comments:

Reviewer #1 (Remarks to the Author):

The authors studied the neuronal coding in the barrel cortex of mice using two transgenic mouse lines (GIN-GFP and GAD-GFP) to investigate the involvement of interneurons in the neurotransmission of cortical excitatory neurons. They injected a viral vector coding for channelrhodopsin and mcherry in the barrel cortex to infect glutamatergic neurons. They recorded GFP cells using ex vivo electrophysiology (patch clamp) in Current clamp mode while optically stimulating ChR2-expressing neurons. They used a one-photon microscope with a digital micromirror device that led to optically stimulated ten areas of interest (ROI) to encode a random 1-bit signal. They generated two different signals: One encoding spike synchrony and the other for spike rate. The 1-bit signal with the 10 ROI induced a polysynaptic activity in the recorded neuron.

Consequently, the response analysis was divided into early (monosynaptic influence, first 5ms) and late response (polysynaptic activity). Interestingly GIN-GFP positive cells and GAD GFP positive cells could integrate the random 1-bit signal independently of the code (synchrony or rate) with nuances.

GAD67+ were more active during the synchrony encoded signal early response than GIN+, while the GIN+ were more active for the rate encoded signal than GAD67+. Late responses were similar between GFP positive cells for synchronicity signal but displayed higher integration than non GFP cells. The average membrane potential of recorded cells were similar.

It is an exciting paper highlighting the notion of sub-network and signal integration in the layer 2/3-barrel cortex of mice. The authors tried to understand the signal parameters that could lead to specific integration in interneurons. The principal component analysis (PCA) brings a powerful insight into the electrophysiological data, reducing the dimensions. The nuances of the code (synchronicity of the rate) reveal another complexity of the cortical network.

The paper is well written.

Comments :

1 – Figure 1: They combine electrophysiological cell properties, PCA, and immunostaining to determine the GFP positive cell identity in the GAD-GFP and GIN-GFP cortical slices. In the GIN-GFP line, the data are straightforward, with almost 100% of GFP cells being Somatostatin positive. It is not the case of the GAD67-GFP lines with 60% of GFP cells being Parvalbumin positive. Based on the analysis and the statistics, they assume the identity of the cells.

I wonder why the authors did not confirm the recorded cells' identity by performing a cell re-trial staining post-recording as it is commonly made especially for PV. Cortical PV cells are a heterogeneous population as not all Fast spiking neurons are PV, and reciprocally, not all PV cells display fast-spiking properties.

2- Injection of the viral vector. It would be appreciated to know the size of the injected area. Does it affect different layers of the cortex? How deep did it go (volume injection)? Can that volume be similar among animals?

The authors said they stayed > 50µm away from the nearest mCherry positive cells during the stimulation to avoid cross stimulation. Did the authors quantify the number of mcherry cells in the 10 ROI for each recorded cell to normalize the response? The numbers of mcherry stimulated cells would modify the late part of the response.

3- I am surprised that the viral vector used (CamKII- ChR2-mCherry , addgene #26975) led to the expression of ChR2 in mcherry negative cells (page 9) as the mcherry is fused to ChR2 in that construct. I wonder if the authors performed immunostaining against mcherry to amplify the signal to verify if it is just low expression or something else.

4- Figure S3, the authors used glutamatergic drugs to block ChR2 receptors in mcherry negative cells. How have the authors targeted the mcherry negative cells expressing ChR2 in order to

record them in electrophysiology? Also, the "n" is missing from the figure. Are the traces examples? or are they the four recorded cells? What is the percentage of mcherry negative cells?

5- The authors encoded a 1-bit signal by alternating one state with a 0 state. The authors do not explain why they chose a random 1-bit signal rather than a more physiological one? What parameters have determined the choice of 1-Bit signal ?

Reviewer #2 (Remarks to the Author):

Prince et al have studied how subtypes of inhibitory neurons can contribute differently to rate and synchrony coding, by using patterned perturbation and intracellular recording of genetically marked neurons. Both the question and the techniques are interesting. The question of whether neurons encode information in their rates or whether the temporal patterns of their responses are also informative is a fundamental question of neuroscience. Recent technological advances - including the possibility to measure from specific subtypes of neurons, and to perturb them with precise spatiotemporal patterns - have provided a unique opportunity to address this longstanding quest. The authors are indeed using all these toolkits combined with an information-theoretic approach to shed a new light on this important question.

The paper argues for the following main finding: that there exists a division of labour between inhibitory neurons, whereby fast-spiking inhibitory neurons are more informative about the temporal pattern of input at an early phase, while regular-spiking interneurons carry more information in later phases during rate encoding. Overall, this result seems to be consistent with a consensus on a division of labour between PV+ and SOM+ interneurons, where the former is tuned for fast/transient responses, while the latter is engaged more slowly and optimized for more integrative computations (also reflected in different patterns of short-term plasticity of the two subtypes documented by recent experiments). The paper is generally well-written and the methodological details are well described and easy to follow.

I therefore think these results are interesting and could be relevant to the general readership of Communications Biology. However, I believe there are several major points to be addressed in order to substantiate the claims. I list them below, with some minor comments coming after.

Major points:

[1] The authors report a reduction in the average spiking activity (and the mean membrane potential) in the synchrony encoding experiments (Fig. 3B and Fig. 4B,C), when the stimulus is off. This is confusing (naturally, one expects the activity to increase when the stimulus is on, even with a temporal code), but they don't explain why or how this may happen.

To address this point, they can discuss possible sources of this effect: for instance, is it related to the fact that mCherry is not only expressed in (presumably) excitatory neurons and some inhibitory neurons might also be infected (Fig. S3) - and that these infected inhibitory neurons could potentially be more sensitive to the synchronous input? Is it related to suppression of variability of pyramidal cells to fluctuating input (Mainen and Sejnowski, Science 1995) - which could also explain lower variability of the activity when synchrony stimulus is off? Or is it related to something more complex at the recurrent level (e.g. more complex recurrent interaction of excitatory-inhibitory neurons, either in the form of oscillations or inhibitory-stabilized networks)? It is important to discuss and clarify this confusing effect and provide a mechanistic understanding of why this happens. Ideally, they can show how this emerges in a simple model (see below, point [3]).

[2] Although the authors make sure to control for the input rate between the two conditions (Fig. 3B,C), there is still a huge difference between the output rates of neurons as a result of different stimulation protocols (Fig. 4 versus Fig. 7 - almost two-fold increase in the rate encoding case for most interneurons/stimulus conditions). This is crucial because high firing rate regimes might show a more linear transformation between membrane potential and spiking activity. This can in turn lead to more information being present in the suprathreshold activity of neurons with higher

spiking activity.

For instance, is the fact that PV neurons (but not others) show more mutual information in their spiking activity in the early stages of the synchrony code (Fig. 5) related to their higher activity at this stage? In other words, the authors should rule out that (or clarify if) simply higher firing rates do not explain their results (see also point [3] below).

A direct analysis of the relationship between mutual information and firing rates should be able to address this. For instance, in Fig. 5C, do the 3 outlier PV neurons with higher MI have higher firing rates too? This is especially interesting to know in view of a bimodal distribution of PV neurons reported in Fig. 4C, middle.

This is an important example as it relates to one of the fundamental claims of the paper. But they can also test this more systematically for other neurons and other conditions. They can do this by analyzing mutual information (MI) as a function of firing rates (similar to what they do for PCs in Fig. 6 and Fig. 9).

To control for the change in output firing rates, currently two different measures of information are used for synchrony (MI) and rate encoding (conditional MI) experiments. This makes the comparison difficult. At least in supplementary figures the other measure should be reported for each condition, respectively. One way to compare the two conditions with the same metric could be to choose a subset of neurons with similar firing rates and compare them - provided enough neurons are available to obtain statistical power.

[3] Related to both previous points is a lack of a mechanistic understanding of why and how different interneurons respond differently. As such, the paper in its current form falls short of providing a convincing explanation of how the main results emerge and which mechanisms are key for the main findings. For instance: is higher spiking activity (either in the transient or later stage) enough to explain the key differences between PV and SOM neurons in synchrony and rate encoding experiments? Would the same neuron model with different thresholds replicate the key findings? Or if other mechanisms like adaptation or even network mechanisms like recurrent interaction of excitation and inhibition are necessary (see point [4] below)?

The authors have provided some tentative explanations themselves, e.g.:

"This effect could be driven by the the faster spiking dynamics and lower membrane time constants of FS GAD67+ interneurons" (L. 326)
but these should be evaluated more directly and clearly.

As the authors have measured or estimated most key physiological parameters of the neurons, it should not be difficult to build single cell models of each subtype and investigate how different patterns of input drive them, and how this changes encoding and MI. It would then be possible to block different mechanisms to see which one is key to reproduce the main results.

The modelling can specifically address main issues raised before and try to explain them:

How can similar membrane potential distributions lead to different spiking behavior (Fig. 4 and Fig. 5)?

Why do we see less spiking activity when stimulus is off in the synchrony encoding protocol (point [1])?

Why is there such a big difference in output firing activity despite similar input rates between the two encoding protocols (point [2])?

[4] Beyond single cell mechanisms, I miss a discussion of other, potentially relevant mechanisms, especially from the network perspective: how may their results be affected by network interactions, especially as they mention the "prolonged polysynaptic activation of neurons" (L. 212) due to lack of inhibition and spatial precision in their stimulation technique?

PV and SOM neurons are known to be involved in prominent oscillations of Gamma/Beta bands (Chen et al., Neuron 2017; Veit et al., Nature Neuroscience 2017), how these oscillations may affect their observations, and how their findings in turn might be linked to the role of interneurons in generating oscillations?

Finally, how can the differential rate/temporal coding of various subtypes revealed here be linked to network level computations that may emerge from - or revealed by - patterned perturbations

(Fan et al., Cell 2020; Sadeh and Clopath, eLife 2020)?

Minor points:

[5] The first part of the results section is dedicated to methodological ways of identifying the subtypes. While this is important, it is long for the results sections (3/10 total pages of results) and distracts the main flow of the paper - this part can be condensed to more quickly transition to the main results and defer the rest of it to the supplements, as often done in similar papers.

[6] It has been argued that the patterned optical illuminations are more "natural" by comparing them to wide-field illumination (L. 181 onwards and Fig. S4). But why is the frequency of the wide-field perturbations much faster (20 Hz)? Doesn't this compromise the comparison?

Also, numbers on x/y-axes of Fig. S4A (logarithmic scales) seem not to be rendered properly.

[7] Legends/figures are not always enough informative, e.g.:

Fig. 4: not clear which color is referring to which cell type (needs referring to other figures)

[8] Typos or rewriting needed:

Fig. 4 and Fig. 7, panel B on x-axis: Milivolt → Millivolt

L. 249: f-I slope

L. 264: except that now the responses during the 0 state (low rate) were lower magnitude and less noisy than ...

L. 266: the 0 state were often hyperpolarized and low variance

L. 307: Together, these [results?] indicate that ...

L. 362: there are likely important differences in interneuron activity in-vivo that relate to things like movement or neuromodulation

I find this last sentence of the discussion a rather strong claim, probably stronger than what could be supported by the evidence provided in the paper:

L. 376: "This suggests that the brain does indeed use both rate and timing codes, and may do so using different mechanisms and for different purposes."

Neocortical inhibitory interneuron subtypes are differentially attuned to synchrony- and rate-coded information

Response to Reviewers

We thank the editors and reviewers for their helpful comments on our manuscript. We were very pleased that the reviewers felt that it is an “exciting paper” and that our results are “...interesting and could be relevant to the general readership of Communications Biology”.

We have made a number of changes to the paper and included new simulation results and analyses which we believe address the reviewers’ concerns and questions. Below, we provide a detailed account of these changes. Please note, editor’s and reviewers’ comments are in blue and any new text is in red (both here, and in the updated manuscript).

Response to Reviewer 1

1. I wonder why the authors did not confirm the recorded cells' identity by performing a cell retrieval staining post-recording as it is commonly made especially for PV. Cortical PV cells are a heterogeneous population as not all Fast spiking neurons are PV, and reciprocally, not all PV cells display fast-spiking properties.

This is a relevant question, as morphological reconstruction would indeed have helped us to further narrow down the identity of our recorded cells. Being completely candid with the reviewer, we did attempt biocytin staining, but we were unsuccessful in reconstructing the morphologies of most of the neurons in our dataset. We worked to trouble-shoot this process for a long time, and managed to get a handful of cells in our GAD67 mice with decent staining (which, qualitatively, indicated that they were basket cells). But, given that most of the cells in our dataset did not have morphological data to confirm specific interneuron identity we chose to emphasise the distinct electrophysiological characteristics of cells in our dataset. We believe that this approach makes sense in our specific case, as we are here investigating electrophysiological features of these cells (e.g. spiking). Of course, ideally, we would have the full morphological information for all the cells, so we hope that future work in this area can make those links even if our present dataset cannot accomplish that. We have inserted some new text into the manuscript discussion to recognize this limitation in our study:

“Third, our dataset did not include morphological information for the neurons, which also helps to determine classifications of interneurons¹⁷. As such, the cells in our study may have actually belonged to subclasses which could have further refined our results. Hopefully, the relationship between morphology and information coding specialization will be explored by follow-up studies.”

- Page 28, *Discussion*

2. Injection of the viral vector. It would be appreciated to know the size of the injected area. Does it affect different layers of the cortex? How deep did it go (volume injection)? Can

that volume similar among animals? The authors said they stayed > 50µm away from the nearest mCherry positive cells during the stimulation to avoid cross stimulation. Did the authors quantify the number of mcherry cells in the 10 ROI for each recorded cell to normalize the response? The numbers of mcherry stimulated cells would modify the late part of the response.

a) *Size of the injected area:*

In the majority of our slices, the spread of the infected area was around 200-250 µm from the injection site, encompassing layers 2/3 and 4. In a minority of slices some viral expression was also evident in superficial layer V. We now state this clearly in the methods:

“Based on the extent of mCherry expression, we typically observed expression up to 200-250 µm from the injection site. This encompassed layer 2/3, but also layer 4, and in a small number of animals, the superficial portion of layer 5.”

- Page 29, *Methods*

b) *Number of mCherry cells in the ROIs:*

We always drew ROIs such that there was only one fluorescent cell visible within the region. Of course, there are two potential issues that prevent us from stating definitively that there was only one ChR2 expressing cell per ROI. First, as noted in the paper and discussed more in the next point, we did observe some ChR2 expression in cells without clear mCherry expression. Second, though we tried to check the entire depth of the slice, it is possible that we occasionally failed to notice cells above or below the focal plane. Thus, although we can say that we only ever saw a single mCherry expressing neuron for each ROI, we must be honest with ourselves and the reviewers that we cannot guarantee that no other ChR2 expressing cells were ever contained in the ROIs. Nonetheless, we do believe that we can say with confidence that there was never more than one mCherry cell per ROI to our knowledge. We now clarify these matters in the manuscript:

“We always drew the ROIs such that only a single mCherry+ neuron could be seen within the ROI. We cannot guarantee that no other cells that expressed ChR2 were ever contained within the ROIs, but we can say that we never observed more than one mCherry+ cell per ROI.”

- Page 31, *Methods*

3. I am surprised that the viral vector used (CamKII- ChR2-mCherry , addgene #26975) led to the expression of ChR2 in mcherry negative cells (page 9) as the mcherry is fused to ChR2 in that construct. I wonder if the authors performed immunostaining against mcherry to amplify the signal to verify if it is just low expression or something else.

We did not try immunostaining against mCherry to amplify the signal. As such, it is indeed possible, per the reviewer’s comment, that when we found ChR2 responses in an apparently mCherry negative cell, it was in fact just due to low expression. In-line with this reasoning,

we note that this may have arisen from the fact that this specific viral vector uses a shortened version of the CamKii promoter, which could lack control sequences that ensure selectivity. This may lead to some low levels of expression in interneurons. We now include comments to note this issue in the manuscript:

“...we observed ChR2 responses in both mCherry+ and mCherry- cells (Fig. S3), **which is in-line with reports that a shortened CamKii promoter can sometimes lead to small amounts of expression beyond pyramidal neurons** ³¹.”

- Page 9, *Results*

4. Figure S3, the authors used glutamatergic drugs to block ChR2 receptors in mcherry negative cells. How have the authors targeted the mcherry negative cells expressing ChR2 in order to record them in electrophysiology? Also, the “n” is missing from the figure. Are the traces examples? or are they the four recorded cells? What is the percentage of mcherry negative cells?

We did not actually target mCherry negative cells that expressed ChR2 in any way to obtain the recordings shown in Figure S3. Instead, these recordings were taken from four randomly selected mCherry negative cells in our slices. We did these recordings precisely because a colleague mentioned to us the issue of lack of selectivity with shortened CamKii promoters (see e.g. Nathanson et al., 2009, *Frontiers in neural circuits*, 3:19), and we wanted to determine whether we were getting any ChR2 expression in apparently mCherry negative cells. After conducting these four recordings, and seeing clear ChR2 responses that could not be prevented with glutamatergic blockade in 2 out of the 4 cells, we were convinced that our colleague was correct and that we were getting some ChR2 expression in mCherry negative cells. We included the data here in Figure S3 precisely because we do not want to mislead any readers regarding the specificity of our viral vector. To clarify our selection process for this data we have updated the figure caption and we have made clear that n = 4.

5. The authors encoded a 1-bit signal by alternating one state with a 0 state. The authors do not explain why they chose a random 1-bit signal rather than a more physiological one? What parameters have determined the choice of 1-Bit signal?

This is an excellent question. In an ideal world we would have used physiological signals as inputs, e.g. whisker deflection sequences. However, mutual information is a notoriously difficult quantity to estimate between signals. One of the major challenges with more biologically realistic signals is that they require exponentially longer experiments to adequately sample from the distribution of responses to obtain a low variance estimate of mutual information, or one must use approximations that can have quite high bias (see, e.g. Paninski, 2003, *Neural Computation*, 15(6): 1191-1253). In contrast, mutual information of a 1-bit signal can be adequately estimated with histograms within realistic experimental time-scales for whole-cell patch clamp, since we can easily obtain a lot of samples from a binary distribution. Thus, we chose a random 1-bit signal because this permitted tractable estimation of mutual information. We note that this still leads to more naturalistic response profiles than, for example, trains of whole-field flashes (**Fig. S4**) However, we now see that

we were not sufficiently clear about this choice in the original manuscript, so we have updated the section to make it as clear as possible:

“We chose a 1-bit random signal because it enables unbiased, low-variance estimation of mutual information with limited samples, unlike more natural, continuous signals”

- Page 11, *Results*

Response to Reviewer 2

1. The authors report a reduction in the average spiking activity (and the mean membrane potential) in the synchrony encoding experiments (Fig. 3B and Fig. 4B,C), when the stimulus is off. This is confusing (naturally, one expects the activity to increase when the stimulus is on, even with a temporal code), but they don't explain why or how this may happen.

This is an important point to clarify, so we are glad that the reviewer has raised this question. First, we would note that the 0 and 1 states do not correspond to off and on states for the stimulus, but rather, to low/high states for the particular coding modality, i.e, low/high rate or low/high synchrony. In the case of rate, the high state does indeed involve more overall activation. But, in the synchrony coding case, the rate of activation (2.7 Hz) is the same in the low and high synchrony states, and so the overall amount of activation is equivalent. Second, in the high synchrony state, all 10 ROIs are activated or none are activated, and when no ROIs are activated, there is zero input to the tissue. As a result, the high synchrony state involves occasional large amounts of activation that easily induce spiking, punctuated by longer periods of no activation and no spiking. This results in a lower average firing rate, since there is little opportunity for depolarization to induce spiking outside of the occasional synchronous set of inputs that induce very reliable spiking. In contrast, in the low synchrony state, there is at least one ROI being activated far more often than in the high synchrony state, leading to fewer periods of total inactivity, and giving far more opportunity for spiking. Thus, the low synchrony state actually drives higher average firing rates. We have now added some clarification to the manuscript to help explain this issue:

“At first glance, these results may be counter-intuitive, since they show that the 0 state of low synchrony induced higher average firing rates. But, careful consideration of the low versus high synchrony states shows that during the low synchrony state there are fewer periods where *no* stimulation occurs, whereas in the high synchrony state stimulation is less frequent, though it is more consistent and strong when it does occur. This leads naturally to higher, though more variable, firing-rates during the low synchrony state.”

- Page 13, *Results*

2. Although the authors make sure to control for the input rate between the two conditions (Fig. 3B,C), there is still a huge difference between the output rates of neurons as a result of different stimulation protocols (Fig. 4 versus Fig. 7 - almost two-fold increase in

the rate encoding case for most interneurons/stimulus conditions). This is crucial because high firing rate regimes might show a more linear transformation between membrane potential and spiking activity. This can in turn lead to more information being present in the suprathreshold activity of neurons with higher spiking activity.

For instance, is the fact that PV neurons (but not others) show more mutual information in their spiking activity in the early stages of the synchrony code (Fig. 5) related to their higher activity at this stage? In other words, the authors should rule out that (or clarify if) simply higher firing rates do not explain their results (see also point [3] below).

A direct analysis of the relationship between mutual information and firing rates should be able to address this. For instance, in Fig. 5C, do the 3 outlier PV neurons with higher MI have higher firing rates too? This is especially interesting to know in view of a bimodal distribution of PV neurons reported in Fig. 4C, middle.

This is an important example as it relates to one of the fundamental claims of the paper. But they can also test this more systematically for other neurons and other conditions. They can do this by analyzing mutual information (MI) as a function of firing rates (similar to what they do for PCs in Fig. 6 and Fig. 9).

This is an excellent point, and very important to consider in our paper. Based on this comment, we have analysed the correlation between mutual information and firing rates in a new Supplementary Figure (**Fig. S8**). Postsynaptic firing rates were computed by counting spikes occurring within the relevant window and averaging across time. We see no significant correlation between mutual information (or conditional mutual information) of the 1-bit signal and the average membrane potential under any conditions (**Fig. S8 A-B, E-F**). But, we do see significant correlations between mutual information (or conditional mutual information) of the 1-bit signal and postsynaptic spike counts, and the postsynaptic firing rate (**Fig. S8 C-D, G-H**). This is to be expected since the two quantities are related, i.e, both are a function of postsynaptic spike counts and so will necessarily be highly correlated.

Unfortunately we do not have sufficient samples to adequately control for this, particularly in the case of the temporal coding condition, where most cells in the population display low MI and low firing rates. However, in the rate coding condition, we can see that there is a dependence on cell type - RS cells with similar firing rate to FS cells have much higher conditional MI, particularly in the late window. Moreover, our new computational modelling (see below) suggests that spike threshold is a significant factor in determining the differences between the cell types that we observed. Thus, we believe that, in fact, one of the *reasons* for the differences we observed between cell types is indeed the typical firing rate of the cells. This is further reinforced by our new computational modelling results (see our response to point 4 below, and **Fig. S7**). We now clarify this matter in the manuscript:

“The importance of spiking properties for determining the mutual information results was further supported by additional analyses demonstrating that the amount of mutual information between the 1-bit signal and the spikes of the neurons, but not their average membrane potential, was strongly correlated with the mean firing-rate of the neurons (Fig. S8). Thus, altogether, our data and modelling results suggest that FS GAD67+ are

better placed to respond to synchrony codes over short time-scales than RS GIN+ interneurons due to their spiking properties.”

- Page 16, *Results*

3. To control for the change in output firing rates, currently two different measures of information are used for synchrony (MI) and rate encoding (conditional MI) experiments. This makes the comparison difficult. At least in supplementary figures the other measure should be reported for each condition, respectively. One way to compare the two conditions with the same metric could be to choose a subset of neurons with similar firing rates and compare them - provided enough neurons are available to obtain statistical power.

This is an important point to clarify. First, we would note that our conditional MI measure does not control for the output firing rates of the recorded neurons. Instead, it controls for the amount of synchrony in the inputs. This was critical for us, as we wanted to measure the MI of the rate code separate from the impact of increased synchrony in high rate conditions. Nonetheless, it is perfectly straightforward for us to include conditional MI for synchrony codes and non-conditional MI for rate codes. We now include those results in two new supplementary figures (**Fig. S9 and S10**). Figure S9 shows that there is almost no mutual information between post-synaptic spikes and the 1-bit signal when conditioned on the ROI activation count. In other words, the ROI activation synchrony explains all of the response entirely. This is to be expected since the two states of the synchrony code have equal rates of activation, and so, conditioning on synchrony should eliminate all of the MI. Figure S10 shows that there is no significant difference in mutual information between the rate coded signal and the postsynaptic spike counts of GAD-67+ and GIN+ interneurons when one uses non-conditioned MI. However, for the rate-coded signal, the ROI activation count is now a confounding variable, since higher input rates also result in higher input synchrony (i.e. coactivation of inputs). Hence, when we perform the non-conditioned MI analysis as we do here, we are conflating the impact of increased rate and increased synchrony. Given that the GAD-67+ cells showed more sensitivity to synchrony, and the GIN+ cells more sensitivity to rate, it is not surprising that they show similar amounts of MI when one allows these two factors to be confounded. However, we note that per our original analysis, when conditioning on the ROI activation count, which controls for the response due to the synchrony of inputs, we show that GIN+ cells encode significantly more information about the rate-coded signal (as shown in **Fig 7**). We hope that the inclusion of the data in Figure S8 & S9 helps to make clear why the conditional MI is the appropriate measure to analyse the relationship between the rate-coded signal and response, whereas unconditional MI is not. We have also expanded on this matter in the text:

“Specifically, when a rate coding system is used, higher rates will inevitably lead to a larger number of ROIs being activated synchronously. As a result, unlike the synchrony code where one can manipulate synchrony while leaving the rate constant, it is impossible to manipulate the rate while leaving the synchrony constant. As such we conditioned our mutual information measure on the ROI activation count in each time bin (see Materials and Methods). This conditioning was important, because without it the mutual information estimated in the responses to the rate code would have included

information that results from synchrony of ROI activation, rather than rate of ROI activation (Fig. S9 & S10).”

- Page 20, *Results*

4. Related to both previous points is a lack of a mechanistic understanding of why and how different interneurons respond differently. As such, the paper in its current form falls short of providing a convincing explanation of how the main results emerge and which mechanisms are key for the main findings. For instance: is higher spiking activity (either in the transient or later stage) enough to explain the key differences between PV and SOM neurons in synchrony and rate encoding experiments? Would the same neuron model with different thresholds replicate the key findings? Or if other mechanisms like adaptation or even network mechanisms like recurrent interaction of excitation and inhibition are necessary (see point [4] below)?

As the authors have measured or estimated most key physiological parameters of the neurons, it should not be difficult to build single cell models of each subtype and investigate how different patterns of input drive them, and how this changes encoding and MI. It would then be possible to block different mechanisms to see which one is key to reproduce the main results.

We are glad that reviewer 2 has raised this point. We thank the reviewer for their suggestions of incorporating a computational model into our manuscript. We believe that this has greatly enhanced the quality of our paper.

In response to this point, we have implemented simulations of our key experiments using adaptive exponential-integrate-and-fire (AdExp) neuron models that test the relative importance of factors such as adaptation, linear integration, and spike threshold in determining the difference in interneuron responses to rate and synchrony codes embedded in our tissue stimulation (**Fig S7**). This type of neuron model was chosen since it is flexible enough to model the firing statistics in these cell types, while also possessing far fewer parameters than a biophysical, conductance-based, Hodgkin-Huxley type model. This enabled us to a) use our recordings of fast-spiking and regular-spiking interneurons shown in Fig 1 (and measurements of key electrophysiological features in Fig S1) to constrain our models, b) perform an *in silico* ablation study by swapping parameters across interneuron types to determine how the relationship between intrinsic properties of neurons affected their sensitivity to rate and synchrony codes used in our *in-vitro* experiments.

Critically, we demonstrate that we are able to reproduce the central finding of the manuscript with our model. Specifically, we show that regular-spiking interneurons show greater sensitivity to information encoded in the rate of input activity compared to fast-spiking interneurons, and fast-spiking interneurons show greater sensitivity to information encoded in the synchrony of input activity compared to regular-spiking interneurons (**Fig S7 C, middle panels**). Furthermore, we also show that we can reverse the sensitivity of regular-spiking interneurons to rate and synchrony codes by swapping the adaptation parameters with fast-spiking interneurons (**Fig S7 C, top panels**). This indicates that adaptation mechanisms are crucial for regular-spiking interneurons' sensitivity to rate codes. Moreover, we show that we can reverse the sensitivity of fast-spiking interneurons to rate

and synchrony codes by swapping both the adaptation and spike-threshold parameters (**Fig S7 C, bottom panels**). This indicates that the higher rheobase and lack of adaptation mechanisms are crucial for fast-spiking interneurons' sensitivity to synchrony codes.

The additional methods can now be found in the text, and we have added the following to the results and discussion:

“Finally, in order to reinforce these analyses with a more concrete, mechanistic understanding, we constructed a computational model of FS and RS interneurons with electrophysiological properties that matched the neurons in our recordings (Fig. S7A; see Materials and Methods). We then ran simulations with these model cells wherein we provided them with inputs that matched our optical activation patterns (Fig. S7B). Interestingly, we found that the models cells exhibited the same phenomena as the real neurons: when using a synchrony encoding, simulated FS neurons carried more information about the 1-bit signal in a short time-window than simulated RS neurons. Moreover, the models allowed us to actively manipulate the parameters in an “ablation” study in order to determine which physiological parameters were most important for inducing this difference in information processing. We found that the most important parameters for inducing the FS versus RS mutual information profiles were spike adaptation and spike threshold (Fig. S7C). Given that the first two principal components in our analyses above included these variables, our data suggest that the spiking properties of these two interneuron types can explain the differences we observed.”

- Pages 15-16, *Results*

“Moreover, our computational model (Fig. S7) and analyses of the relationship between firing-rate and mutual information (Fig. S10), further support the idea that differences in spiking properties can explain the differences we observed between FS and RS cell types.”

- Page 27, *Discussion*

5. Beyond single cell mechanisms, I miss a discussion of other, potentially relevant mechanisms, especially from the network perspective: how may their results be affected by network interactions, especially as they mention the “prolonged polysynaptic activation of neurons” (L. 212) due to lack of inhibition and spatial precision in their stimulation technique?
6. PV and SOM neurons are known to be involved in prominent oscillations of Gamma/Beta bands (Chen et al., Neuron 2017; Veit et al., Nature Neuroscience 2017), how these oscillations may affect their observations, and how their findings in turn might be linked to the role of interneurons in generating oscillations?

We have now added some comments about network mechanisms and oscillations in the discussion section:

“One aspect of interneuron specialization for information coding that we have not explored in this paper is the role of network-level interactions between interneuron classes. For example, it is known that PV+ and SST+ interneurons can be connected to one another²⁸, which would mean that the coding properties of one neuron type could affect the other directly. This may have been the case in our data here, given that we could not guarantee that polysynaptic interactions were not involved in generating some of the information processing we observed. However, our computational model (Fig. S7) suggests that at least the core aspects of the information coding specialization observed here can be explained in a cell-autonomous manner. Nonetheless, network interactions are an important consideration here, especially when we consider the potential role of oscillations in shaping synchrony and rate across a network, and the role of interneurons in regulating oscillations⁴³.”

- Pages 27-28, *Discussion*

7. Finally, how can the differential rate/temporal coding of various subtypes revealed here be linked to network level computations that may emerge from - or revealed by - patterned perturbations (Fan et al., Cell 2020; Sadeh and Clopath, eLife 2020)?

Thank you for directing our attention to these papers, they are very interesting, and certainly relevant to the general approach we have taken here. We have now added a comment to this effect in the discussion:

“Such investigations would ideally be done using similar patterned optical activation approaches to ours, but *in vivo*, as data suggests that patterned optical activation can reveal aspects of network organization and function that other forms of stimulation cannot, particularly with respect to inhibitory microcircuits^{41,42}.”

- Page 27, *Discussion*

8. The first part of the results section is dedicated to methodological ways of identifying the subtypes. While this is important, it is long for the results sections (3/10 total pages of results) and distracts the main flow of the paper - this part can be condensed to more quickly transition to the main results and defer the rest of it to the supplements, as often done in similar papers.

We appreciate the reviewer’s perspective here, but we would note that this may be a matter of personal preference for a reader. Indeed, our other reviewer asked many questions about the subtype identification, which suggests that for some readers this is a matter of central interest. With all due respect to this reviewer’s opinion on the matter (which we are actually quite sympathetic to), we have left the length of the text in the first section of the results as is. If the reviewer feels strongly about this point we would be happy to revisit it.

9. It has been argued that the patterned optical illuminations are more “natural” by comparing them to wide-field illumination (L. 181 onwards and Fig. S4). But why is the frequency of the wide-field perturbations much faster (20 Hz)? Doesn’t this compromise the comparison? Also, numbers on x/y-axes of Fig. S4A (logarithmic scales) seem not to be rendered properly.

With regards to the whole-field illumination, the frequency of an individual ROI in our protocols is slower than 20 Hz, but the frequency of optical stimulation is actually closer to 20 Hz for the high rate coding state (because the ROIs activate at different times). Thus, we were attempting to match the optical rate of activation, not the ROI rate of activation. With regards to the legend, we have corrected the improper rendering of Fig S4A, thank you for catching this.

10. Legends/figures are not always enough informative, e.g.: Fig. 4: not clear which color is referring to which cell type (needs referring to other figures)

This is a good point, we have added color labels for all the figures now.

11. Typos or rewriting needed

Thank you for pointing out a number of typos. We have since corrected them.

REVIEWERS' COMMENTS:

Reviewer #1 (Remarks to the Author):

I would like to thank the authors for the quality of the answers. They detailed with honesty point-by-point responses to all comments. The manuscript is more precise than before, highlighting the fineness of the work done.

Reviewer #2 (Remarks to the Author):

The new modelling, analyses and explanations (especially inclusion of computational modelling - Fig. S7) have improved the manuscript and addressed my concerns, so I don't have any further comment.

Typo: Line 649 $\tau_e = 5 \text{ ms}$  $\tau_e = 5 \text{ ms}$

Neocortical inhibitory interneuron subtypes are differentially attuned to synchrony- and rate-coded information

Response to Reviewers

We thank the reviewers for their recommendation to accept our paper. Their comments and insights have greatly enhanced the manuscript and we are pleased to submit a final revision to *Communications Biology*.